# Leveraging Predictions in Smoothed Online Convex Optimization via Gradient-based Algorithms

**Yingying Li**
SEAS
Harvard University
Cambridge, MA, USA. 02138
yingyingli@g.harvard.edu

**Na Li**
SEAS
Harvard University
Cambridge, MA, USA. 02138
nali@seas.harvard.edu

## Abstract

We consider online convex optimization with time-varying stage costs and additional switching costs. Since the switching costs introduce coupling across all stages, multi-step-ahead (long-term) predictions are incorporated to improve the online performance. However, longer-term predictions tend to suffer from lower quality. Thus, a critical question is: *how to reduce the impact of long-term prediction errors on the online performance?* To address this question, we introduce a gradient-based online algorithm, Receding Horizon Inexact Gradient (RHIG), and analyze its performance by dynamic regrets in terms of the temporal variation of the environment and the prediction errors. RHIG only considers at most $W$-step-ahead predictions to avoid being misled by worse predictions in the longer term. The optimal choice of $W$ suggested by our regret bounds depends on the tradeoff between the variation of the environment and the prediction accuracy. Additionally, we apply RHIG to a well-established stochastic prediction error model and provide expected regret and concentration bounds under correlated prediction errors. Lastly, we numerically test the performance of RHIG on quadrotor tracking problems.

## 1 Introduction

In this paper, we consider online convex optimization (OCO) with switching costs, also known as "smoothed" OCO (SOCO) in the literature [1–5]. The stage costs are time-varying but the decision maker (agent) has access to noisy predictions on the future costs. Specifically, we consider stage cost function $f(x_t; \theta_t)$ parameterized by a time-varying parameter $\theta_t \in \Theta$. At each stage $t \in \{1, 2, \cdots, T\}$, the agent receives the predictions of the future parameters $\theta_{t|t-1}, \ldots, \theta_{T|t-1}$, takes an action $x_t \in \mathbb{X}$, and suffers the stage cost $f(x_t; \theta_t)$ plus a switching cost $d(x_t, x_{t-1})$. The switching cost $d(x_t, x_{t-1})$ penalizes the changes in the actions between consecutive stages. This problem enjoys a wide range of applications. For example, in the data center management problems [6, 7], the switching cost captures the switch on/off costs of the servers [7], and noisy predictions on future electricity prices and network traffic are available for the center manager [8, 9]. Other applications include smart building [10, 11], robotics [12], smart grid [13], connected vehicles [14], optimal control [15], etc.

Unlike OCO [16], the switching costs considered in SOCO introduce coupling among all stages, so multi-step-ahead predictions are usually used for promoting the online performance. However, in most cases, predictions are not accurate, and longer-term predictions tend to suffer lower quality. Therefore, it is crucial to study *how to use the multi-step-ahead predictions effectively*, especially, *how to reduce the impact of long-term prediction errors on the online performance.*

Recent years have witnessed a growing interest in studying SOCO with predictions. However, most literature avoids the complicated analysis on noisy multi-step-ahead predictions by considering a

rather simplified prediction model: the costs in the next $W$ stages are accurately predicted with no errors while the costs beyond the next $W$ stages are adversarial and not predictable at all [3,6,7,15,17]. This first-accurate-then-adversarial model is motivated by the fact that long-term predictions are much worse than the short-term ones, but it fails to capture the gradually increasing prediction errors as one predicts further into the future. Several online algorithms have been proposed for this model, e.g. the optimization-based algorithm AFHC [6], the gradient-based algorithm RHGD [17], etc. Moreover, there have been a few attempts to consider noisy multi-step-ahead predictions in SOCO. In particular, [1] proposes a stochastic prediction error model to describe the correlation among prediction errors. This stochastic model generalizes stochastic filter prediction errors. Later, [18] proposes an optimization-based algorithm CHC, which generalizes AFHC and MPC [19], and analyzes its performance based on the stochastic model in [1].

However, many important questions remain unresolved for SOCO with noisy predictions. For example, though the discussions on the stochastic model in [1, 18] are insightful, there still lacks a general understanding on the effects of prediction errors on SOCO without any (stochastic model) assumptions. Moreover, most methods in the literature [1, 6, 18] require fully solving multi-stage optimization programs at each stage; it is unclear whether any gradient-based algorithm, which is more computationally efficient, would work for SOCO with noisy multi-step-ahead predictions.

**Our contributions.** In this paper, we introduce a gradient-based online algorithm Receding Horizon Inexact Gradient (RHIG). It is a straightforward extension of RHGD, which was designed for the simple first-accurate-then-adversarial prediction model in [17]. In RHIG, the agent can choose to utilize only $W \geq 0$ steps of future predictions, where $W$ is a tunable parameter for the agent.

We first analyze the dynamic regret of RHIG by considering general prediction errors without any (stochastic model) assumptions. Our regret bound depends on both the errors of the utilized predictions, i.e. $k$-step-ahead prediction errors for $k \leq W$; and the temporal variation of the environment $V_T = \sum_{t=1}^{T} \sup_{x \in \mathbb{X}} |f(x; \theta_t) - f(x; \theta_{t-1})|$. Interestingly, the regret bound shows that the optimal choice of $W$ depends on the tradeoff between the variation of environment $V_T$ and the prediction errors, that is, a large $W$ is preferred when $V_T$ is large while a small $W$ is preferred when the prediction errors are large. Further, the $k$-step prediction errors have an exponentially decaying influence on the regret bound as $k$ increases, indicating that RHIG effectively reduces the negative impact of the noisy multi-step-ahead predictions.

We then consider the stochastic prediction error model in [1, 18] to analyze the performance of RHIG under correlated prediction errors. We provide an expected regret bound and a concentration bound on the regret. In both bounds, the long-term correlation among prediction errors has an exponentially decaying effect, indicating RHIG's good performance even with strongly correlated prediction errors.

Finally, we numerically test RHIG on online quadrotor tracking problems. Numerical experiments show that RHIG outperforms AFHC and CHC especially under larger prediction errors. Besides, we show that RHIG is robust to unforeseen shocks in the future.

**Additional related work:** There is a related line of work on predictable OCO (without switching costs) [20–23]. In this case, stage decisions are fully decoupled and only one-step-ahead predictions are relevant. The proposed algorithms include OMD [21, 22], DMD [20], AOMD [23], whose regret bounds depend on one-step prediction errors [20, 22, 23] and $V_T$ if dynamic regret is concerned [23].

Besides, it is worth mentioning the related online decision making problems with coupling across stages, e.g. OCO with memory [24, 25], online optimal control [15, 26–28], online Markov decision processes [29–31], etc. Leveraging inaccurate predictions in these problems is also worth exploring.

**Notation:** $\Pi_{\mathbb{X}}$ denotes the projection onto set $\mathbb{X}$. $\mathbb{X}^T = \mathbb{X} \times \cdots \times \mathbb{X}$ is a Cartesian product. $\nabla_x$ denotes the gradient with $x$. $\sum_{t=0}^{k} a_t = 0$ if $k < 0$. $\| \cdot \|_F$ and $\| \cdot \|$ are Frobenius norm and $L_2$ norm. $\mathbb{1}_E$ denotes an indicator function on set $E$.

## 2   Problem Formulation

Consider stage cost function $f(x_t; \theta_t)$ with a time-varying parameter $\theta_t \in \Theta$ and a switching cost $d(x_t, x_{t-1})$ that penalize the changes in the actions between stages. The total cost in horizon $T$ is: $C(\boldsymbol{x}; \boldsymbol{\theta}) = \sum_{t=1}^{T} [f(x_t; \theta_t) + d(x_t, x_{t-1})]$, where $x_t \in \mathbb{X} \subseteq \mathbb{R}^n$, $\theta_t \in \Theta \subseteq \mathbb{R}^p$, and we denote $\boldsymbol{x} := (x_1^\top, \ldots, x_T^\top)^\top$, $\boldsymbol{\theta} = (\theta_1^\top, \ldots, \theta_T^\top)^\top$. The switching cost enjoys many applications as discussed

in Section 1. The presence of switching costs $d(x_t, x_{t-1})$ couples decisions among stages. Therefore, all parameters in horizon $T$, i.e. $\theta_1, \ldots, \theta_T$, are needed to minimize $C(\boldsymbol{x}; \boldsymbol{\theta})$. However, in practice, only predictions are available ahead of the time and the predictions are often inaccurate, especially the long-term predictions. This may lead to wrong decisions and degrade the online performance. In this paper, we aim at designing an online algorithm to use prediction effectively and unveil the unavoidable influences of the prediction errors on the online performance.

**Prediction models.** In this paper, we denote the prediction of the future parameter $\theta_\tau$ obtained at the beginning of stage $t$ as $\theta_{\tau|t-1}$ for $t \leq \tau \leq T$. The initial predictions $\theta_{1|0}, \ldots, \theta_{T|0}$ are usually available before the problem starts. We call $\theta_{t|t-k}$ as $k$-step-ahead predictions of parameter $\theta_t$ and let $\delta_t(k)$ denote the $k$-step-prediction error, i.e.

$$\delta_t(k) := \theta_t - \theta_{t|t-k}, \quad \forall 1 \leq k \leq t. \tag{1}$$

For notation simplicity, we define $\theta_{t|\tau} := \theta_{t|0}$ for $\tau \leq 0$, and thus $\delta_t(k) = \delta_t(t)$ for $k \geq t$. Further, we denote the vector of $k$-step prediction errors of all stages as follows

$$\boldsymbol{\delta}(k) = (\delta_1(k)^\top, \ldots, \delta_T(k)^\top)^\top \in \mathbb{R}^{pT}, \qquad \forall 1 \leq k \leq T. \tag{2}$$

It is commonly observed that the number of lookahead steps heavily influences the prediction accuracy and in most cases long-term prediction errors are usually larger than short-term ones.

We will first consider the general prediction errors without additional assumptions on $\delta_t(k)$. Then, we will carry out a more insightful discussion for the case when the prediction error $\|\delta_t(k)\|$ is non-decreasing with the number of look-ahead steps $k$. Further, it is also commonly observed that the prediction errors are correlated. To study how the correlation among prediction errors affect the algorithm performance, we adopt the stochastic model of prediction errors in [1]. The stochastic model is a more general version of the prediction errors for Wiener filter, Kalman filter, etc. In Section 5, we will review this stochastic model and analyze the performance under this model.

**Protocols.** We summarize the protocols of our online problem below. We consider that the agent knows the function form $f(\cdot; \cdot)$ and $d(\cdot, \cdot)$ a priori. For each stage $t = 1, 2, \ldots, T$, the agent

- receives the predictions $\theta_{t|t-1}, \ldots, \theta_{T|t-1}$ at the beginning of stage;[1]
- selects $x_t$ based on the predictions and the history, i.e. $\theta_1, \ldots, \theta_{t-1}, \theta_{t|t-1}, \ldots, \theta_{T|t-1}$;
- suffers $f(x_t; \theta_t) + d(x_t, x_{t-1})$ at the end of stage after true $\theta_t$ is revealed.

**Performance metrics.** This paper considers (expected) dynamic regret [23]. The benchmark is the optimal solution $\boldsymbol{x}^*$ in hindsight when $\boldsymbol{\theta}$ is known, i.e. $\boldsymbol{x}^* = \arg\min_{\boldsymbol{x} \in \mathbb{X}^T} C(\boldsymbol{x}; \boldsymbol{\theta})$, where $\boldsymbol{x}^* = ((x_1^*)^\top, \ldots, (x_T^*)^\top)^\top$. Notice that $\boldsymbol{x}^*$ depends on $\boldsymbol{\theta}$ but we omit $\boldsymbol{\theta}$ for brevity. Let $\boldsymbol{x}^{\mathcal{A}}$ denote the actions selected by the online algorithm $\mathcal{A}$. The dynamic regret of $\mathcal{A}$ with parameter $\boldsymbol{\theta}$ is defined as

$$\text{Reg}(\mathcal{A}) = C(\boldsymbol{x}^{\mathcal{A}}; \boldsymbol{\theta}) - C(\boldsymbol{x}^*; \boldsymbol{\theta}) \tag{3}$$

When considering stochastic prediction errors, we define the expectation of the dynamic regret: $\mathbb{E}[\text{Reg}(\mathcal{A})] = \mathbb{E}\left[C(\boldsymbol{x}^{\mathcal{A}}; \boldsymbol{\theta}) - C(\boldsymbol{x}^*; \boldsymbol{\theta})\right]$, where the expectation is taken with respect to the randomness of the prediction error as well as the randomness of $\theta_t$ if applicable.

Lastly, we consider the following assumptions throughout this paper.

**Assumption 1.** $f(x; \theta)$ is $\alpha$ strongly convex and $l_f$ smooth with respect to $x \in \mathbb{X}$ for any $\theta \in \Theta$. $d(x, x')$ is convex and $l_d$ smooth with respect to $x, x' \in \mathbb{X}$.

**Assumption 2.** $\nabla_x f(x; \theta)$ is $h$-Lipschitz continuous with respect to $\theta$ for any $x$, i.e.

$$\|\nabla_x f(x; \theta_1) - \nabla_x f(x; \theta_2)\| \leq h\|\theta_1 - \theta_2\|, \ \forall x \in \mathbb{X}, \ \theta_1, \theta_2 \in \Theta.$$

Assumption 1 is common in convex optimization literature [32]. Assumption 2 ensures a small prediction error on $\theta$ only causes a small error in the gradient. Without such an assumption, little can be achieved with noisy predictions. Lastly, we note that these assumptions are for the purpose of theoretical regret analysis. The designed algorithm would apply for general convex smooth functions.

# 3 Receding Horizon Inexact Gradient (RHIG)

This section introduces our online algorithm Receding Horizon Inexact Gradient (RHIG). It is based on a promising online algorithm RHGD [17] designed for an over-simplified prediction model: at stage $t$, the next $W$-stage parameters $\{\theta_\tau\}_{\tau=t}^{t+W-1}$ are exactly known but parameters beyond $W$ steps are adversarial and totally unknown. We will first briefly review RHGD and then introduce our RHIG as an extension of RHGD to handle the inaccurate multi-step-ahead predictions.

## 3.1 Preliminary: RHGD with accurate lookahead window

RHGD is built on the following observation: the $k$-th iteration of offline gradient descent (GD) on the total cost $C(\boldsymbol{x}; \boldsymbol{\theta})$ for stage variable $x_\tau(k)$, i.e.,

$$x_\tau(k) = \Pi_{\mathbb{X}}[x_\tau(k-1) - \eta \nabla_{x_\tau} C(\boldsymbol{x}(k-1); \boldsymbol{\theta})], \quad \forall 1 \leq \tau \leq T,$$
$$\text{where} \quad \nabla_{x_\tau} C(\boldsymbol{x}; \boldsymbol{\theta}) = \nabla_{x_\tau} f(x_\tau; \theta_\tau) + \nabla_{x_\tau} d(x_\tau, x_{\tau-1}) + \nabla_{x_\tau} d(x_{\tau+1}, x_\tau) \mathbb{1}_{(\tau \leq T-1)}, \tag{4}$$

only requires neighboring stage variables $x_{\tau-1}(k-1), x_\tau(k-1), x_{\tau+1}(k-1)$ and local parameter $\theta_\tau$, instead of all variables $\boldsymbol{x}(k-1)$ and all parameters $\boldsymbol{\theta}$. This observation allows RHGD [17] (Algorithm 1) to implement the offline gradient (4) for $W$ iterations by only using $\{\theta_\tau\}_{\tau=t}^{t+W-1}$. Specifically, at stage $2 - W \leq t \leq T$, RHGD initializes $x_{t+W}(0)$ by an oracle $\phi$ (Line 4), where $\phi$ can be OCO algorithms (e.g. OGD, OMD [16]) that compute $x_{t+W}(0)$ with $\{\theta_t\}_{t=1}^{t+W-1}$.[2] If $t + W > T$, skip this step. Next, RHGD applies the offline GD (4) to compute $x_{t+W-1}(1), x_{t+W-2}(2), \ldots, x_t(W)$, which only uses $\theta_{t+W-1}, \ldots, \theta_t$ respectively (Line 5-7). RHGD skips $x_\tau$ if $\tau \notin \{1, \ldots, T\}$. Finally, RHGD outputs $x_t(W)$, the $W$-th update of offline GD.

---

**Algorithm 1:** Receding Horizon Gradient Descent (RHGD) [17]

---

1: **Inputs:** Initial decision $x_0$; stepsize $\eta$; initialization oracle $\phi$
2: Let $x_1(0) = x_0$.
3: **for** $t = 2 - W, \ldots, T$ **do**
4:     Initialize $x_{t+W}(0)$ by oracle $\phi$ if $t + W \leq T$.
5:     **for** $\tau = \min(t + W - 1, T)$ **downto** $\max(t, 1)$ **do**
6:         Update $x_\tau(t + W - \tau)$ by the offline GD on $x_\tau$ in (4).
7:     Output $x_t(W)$ when $1 \leq t \leq T$.

---

## 3.2 Our algorithm: RHIG for inaccurate predictions

---

**Algorithm 2:** Receding Horizon Inexact Gradient (RHIG)

---

1: **Inputs:** The length of the lookahead horizon: $W \geq 0$; initial decision $x_0$; stepsize $\eta$; initialization oracle $\phi$
2: Let $x_1(0) = x_0$.
3: **for** $t = 2 - W$ to $T$ **do**
4:     **if** $t + W \leq T$ **then**
5:         Compute $x_{t+W}(0)$ by the initialization oracle $\phi$ with inexact information.
6:     **for** $\tau = \min(t + W - 1, T)$ **downto** $\max(t, 1)$ **do**
7:         Compute $x_\tau(t + W - \tau)$ based on the prediction $\theta_{\tau|t-1}$ and the inexact partial gradient:

$$x_\tau(k) = \Pi_{\mathbb{X}}[x_\tau(k-1) - \eta g_\tau(x_{\tau-1:\tau+1}(k-1); \theta_{\tau|t-1})], \quad \text{where } k = t + W - \tau. \tag{5}$$

8:     Output the decision $x_t(W)$ when $1 \leq t \leq T$.

---

With noisy predictions, it is natural to use the prediction $\theta_{\tau|t-1}$ to estimate the future partial gradients,

$$g_\tau(x_{\tau-1:\tau+1}; \theta_{\tau|t-1}) = \nabla_{x_\tau} f(x_\tau; \theta_{\tau|t-1}) + \nabla_{x_\tau} d(x_\tau, x_{\tau-1}) + \nabla_{x_\tau} d(x_{\tau+1}, x_\tau) \mathbb{1}_{(\tau \leq T-1)},$$

and then updates $x_\tau$ by the estimated gradients. This motivates Receding Horizon Inexact Gradient (RHIG) in Algorithm 2. Compared with RHGD, RHIG has the following major differences.

| $x_1(0) = x_0$ | $x_2(0); \phi$ | $x_3(0); \phi$ | $x_4(0); \phi$ | $t = -1$ |
| $x_1(1); \theta_{1\mid-1}$ | $x_2(1); \theta_{2\mid0}$ | $x_3(1); \theta_{3\mid1}$ | $x_4(1); \theta_{4\mid2}$ | $t = 0$ $t = 1$ $t = 2$ |
| $x_1(2); \theta_{1\mid0}$ | $x_2(2); \theta_{2\mid1}$ | $x_3(2); \theta_{3\mid2}$ | $x_4(2); \theta_{4\mid3}$ | $t = 3$ $t = 4$ |

Figure 1: Example: RHIG for $W = 2, T = 4$. (Orange) at $t = -1$, let $x_1(0) = x_0$. (Yellow) at $t = 0$, initialize $x_2(0)$ by $\phi$, then compute $x_1(1)$ by inexact offline GD (5) with prediction $\theta_{1\mid-1} = \theta_{1\mid0}$. (Green) At $t = 1$, initialize $x_3(0)$ by $\phi$, and update $x_2(1)$ and $x_1(2)$ by (5) with $\theta_{2\mid0}$ and $\theta_{1\mid0}$ respectively. At $t = 2$, initialize $x_4(0)$ by $\phi$, then update $x_3(1)$, $x_2(2)$ by inexact offline GD (5) with $\theta_{3\mid1}$ and $\theta_{2\mid1}$ respectively. $t = 3, 4$ are similar. Notice that $\boldsymbol{x}(1) = (x_1(1), \dots, x_4(1))$ is computed by inexact offline gradient with 2-step-ahead predictions, and $\boldsymbol{x}(2)$ by 1-step-ahead predictions.

- (Line 1) Unlike RHGD, the lookahead horizon length $W \geq 0$ is tunable in RHIG. When selecting $W = 0$, RHIG does not use any predictions in Line 5-7. When selecting $1 \leq W \leq T$, RHIG utilizes at most $W$-step-ahead predictions $\{\theta_{\tau\mid t-1}\}_{\tau=t}^{t+W-1}$ in Line 5-7. Specifically, when $W = T$, RHIG utilizes all the future predictions $\{\theta_{\tau\mid t-1}\}_{\tau=t}^{T}$. Interestingly, one can also select $W > T$. In this case, RHIG not only utilizes all the predictions but also conducts more computation based on the initial predictions $\{\theta_{\tau\mid0}\}_{\tau=1}^{T}$ at $t \leq 0$ (recall that $\theta_{\tau\mid t-1} = \theta_{\tau\mid0}$ when $t \leq 0$). Notably, when $W \to +\infty$, RHIG essentially solves $\arg\min_{\boldsymbol{x} \in \mathbb{X}^T} C(\boldsymbol{x}; \{\theta_{\tau\mid0}\}_{\tau=1}^{T})$ at $t \leq 0$ to serve as warm starts at $t = 1$.[3] The choice of $W$ will be discussed in Section 4-5.

- (Line 5) Notice that the oracle $\phi$ no longer receives $\theta_{t+W-1}$ exactly in RHIG, so OCO algorithms need to be modified here. For example, OGD initializes $x_{t+W}(0)$ by prediction $\theta_{t+W-1\mid t-1}$:

$$x_\tau(0) = \Pi_{\mathbb{X}}[x_{\tau-1}(0) - \xi_\tau \nabla_{x_{\tau-1}} f(x_{\tau-1}(0); \theta_{\tau-1\mid t-1})], \quad \text{where } \tau = t + W. \quad (6)$$

Besides, we note that since $\theta_{\tau\mid t-1}$ is available, OGD (6) can also use $\theta_{\tau\mid t-1}$ to update $x_\tau(0)$. Similarly, OCO algorithms with predictions, e.g. (A)OMD [21, 23], DMD [33], can be applied.

- (Line 7) Instead of exact offline GD in RHGD, RHIG can be interpreted as inexact offline GD with prediction errors. Especially, (5) can be written as $x_\tau(k) = x_\tau(k-1) - \eta \nabla_{x_\tau} C(\boldsymbol{x}(k-1); \boldsymbol{\theta} - \boldsymbol{\delta}(W - k + 1))$ by the definition (1). More compactly, we can write RHIG updates as

$$\boldsymbol{x}(k) = \Pi_{\mathbb{X}^T} \left[ \boldsymbol{x}(k-1) - \eta \nabla_{\boldsymbol{x}} C(\boldsymbol{x}(k-1); \boldsymbol{\theta} - \boldsymbol{\delta}(W - k + 1)) \right], \quad \forall 1 \leq k \leq W, \quad (7)$$

where $\nabla_{\boldsymbol{x}} C(\boldsymbol{x}(k-1); \boldsymbol{\theta} - \boldsymbol{\delta}(W - k + 1))$ is an inexact version of the gradient $\nabla_{\boldsymbol{x}} C(\boldsymbol{x}(k-1); \boldsymbol{\theta})$.

Though the design of RHIG is rather straightforward, both theoretical analysis and numerical experiments show promising performance of RHIG even under poor long-term predictions (Section 4-6). Some intuitions are discussed below. By formula (7), as the iteration number $k$ increases, RHIG employs inexact gradients with shorter-term prediction errors $\boldsymbol{\delta}(W - k + 1)$. Since shorter-term predictions are often more accurate than the longer-term ones, RHIG gradually utilizes more accurate gradient information as iterations go on, reducing the optimality gap caused by inexact gradients. Further, the longer-term prediction errors used at the first several iterations are compressed by later gradient updates, especially for strongly convex costs where GD enjoys certain contraction property.

Lastly, with a gradient-based $\phi$ and a finite $W$, RHIG only utilizes gradient updates at each $t$ and is thus more computationally efficient than AFHC [1] and CHC [18] that solve multi-stage optimization.

## 4 General Regret Analysis

This section considers general prediction errors *without* stochastic model assumptions and provides dynamic regret bounds and discussions,[4] before which is a helping lemma on the properties of $C(\boldsymbol{x}; \boldsymbol{\theta})$.

**Lemma 1.** $C(\boldsymbol{x}; \boldsymbol{\theta})$ *is* $\alpha$ *strongly convex and* $L = l_f + 2l_d$ *smooth with* $\boldsymbol{x} \in \mathbb{X}^T$ *for any* $\boldsymbol{\theta} \in \Theta^T$.

The following theorem provides a general regret bound for RHIG with any initialization oracle $\phi$.

**Theorem 1** (General Regret Bound). *Under Assumption 1-2, for $W \geq 0$, oracle $\phi$, $\eta = \frac{1}{2L}$, we have*

$$\text{Reg}(RHIG) \leq \frac{2L}{\alpha}\rho^W \text{Reg}(\phi) + \zeta \sum_{k=1}^{\min(W,T)} \rho^{k-1}\|\boldsymbol{\delta}(k)\|^2 + \mathbb{1}_{(W>T)}\frac{\rho^T - \rho^W}{1-\rho}\zeta\|\boldsymbol{\delta}(T)\|^2, \quad (8)$$

*where $\rho = 1 - \frac{\alpha}{4L}$, $\zeta = \frac{h^2}{\alpha} + \frac{h^2}{2L}$, $\text{Reg}(\phi) = C(\boldsymbol{x}(0); \boldsymbol{\theta}) - C(\boldsymbol{x}^*; \boldsymbol{\theta})$ and $\boldsymbol{x}(0)$ is computed by $\phi$.*

The regret bound (8) consists of three terms. The first term $\frac{2L}{\alpha}\rho^W\text{Reg}(\phi)$ depends on $\phi$. The second term $\zeta \sum_{k=1}^{\min(W,T)} \rho^{k-1}\|\boldsymbol{\delta}(k)\|^2$ and the third term $\mathbb{1}_{(W>T)}\frac{\rho^T-\rho^W}{1-\rho}\zeta\|\boldsymbol{\delta}(T)\|^2$ depend on the errors of the predictions used in Algorithm 2 (Line 5-7). Specifically, when $W \leq T$, at most $W$-step-ahead predictions are used, so the second term involves at most $W$-step-ahead prediction errors $\{\boldsymbol{\delta}(k)\}_{k=1}^W$ (the third term is irrelevant). When $W > T$, RHIG uses all predictions, so the second term includes all prediction errors $\{\boldsymbol{\delta}(k)\}_{k=1}^T$; besides, RHIG conducts more computation by the initial predictions $\{\theta_{t|0}\}_{t=1}^T$ at $t \leq 0$ (see Section 3), causing the third term on the initial prediction error $\|\boldsymbol{\delta}(T)\|^2$.

**An example of $\phi$: restarted OGD [34].** For more concrete discussions on the regret bound, we consider a specific $\phi$, restarted OGD [34], as reviewed below. Consider an epoch size $\Delta$ and divide $T$ stages into $\lceil T/\Delta \rceil$ epochs with size $\Delta$. In each epoch $k$, restart OGD (6) and let $\xi_t = \frac{4}{\alpha j}$ at $t = k\Delta + j$ for $1 \leq j \leq \Delta$. Similar to [34], we define the variation of the environment as $V_T = \sum_{t=1}^T \sup_{x \in \mathbb{X}} |f(x; \theta_t) - f(x; \theta_{t-1})|$, and consider $V_T$ is known and $1 \leq V_T \leq T$.[5] To obtain a meaningful regret bound, we impose Assumption 3, where condition i) is common in OCO literature [23, 34, 35] and condition ii) requires a small switching cost under a small change of actions.

**Assumption 3.** *i) There exists $G > 0$ such that $\|\nabla_x f(x; \theta)\| \leq G$, $\forall x \in \mathbb{X}, \theta \in \Theta$. ii) There exists $\beta$ such that $0 \leq d(x, x') \leq \frac{\beta}{2}\|x - x'\|^2$.[6]*

**Theorem 2** (Regret bound of restarted OGD). *Under Assumption 1-3, consider $T > 2$ and $\Delta = \lceil \sqrt{2T/V_T} \rceil$, the initialization based on restarted OGD described above satisfies the regret bound:*

$$\text{Reg}(OGD) \leq C_1 \sqrt{V_T T} \log(1 + \sqrt{T/V_T}) + \frac{h^2}{\alpha}\|\boldsymbol{\delta}(\min(W, T))\|^2, \quad (9)$$

*where $C_1 = \frac{4\sqrt{2}G^2}{\alpha} + \frac{32\sqrt{2}\beta G^2}{\alpha^2} + 20$.*

Notice that restarted OGD's regret bound (9) consists of two terms: the first term $C_1\sqrt{V_T T}\log(1 + \sqrt{T/V_T})$ is consistent with the original regret bound in [34] for strongly convex costs, which increases with the environment's variation $V_T$; the second term depends on the $\min(W, T)$-step prediction error, which is intuitive since OGD (6) in our setting only has access to the inexact gradient $\nabla_{x_{s-1}} f(x_{s-1}(0); \theta_{s-1|s-W-1})$ predicted by the $\min(W, T)$-step-ahead prediction $\theta_{s-1|s-W-1}$.[7]

**Corollary 1** (RHIG with restarted OGD initialization). *Under the conditions in Theorem 1 and 2, RHIG with $\phi$ based on restarted OGD satisfies*

$$\text{Reg}(RHIG) \leq \underbrace{\rho^W \frac{2L}{\alpha}C_1\sqrt{V_T T}\log(1 + \sqrt{T/V_T})}_{\text{Part I}}$$

$$+ \underbrace{\frac{2L}{\alpha}\frac{h^2}{\alpha}\rho^W\|\boldsymbol{\delta}(\min(W,T))\|^2 + \sum_{k=1}^{\min(W,T)}\zeta\rho^{k-1}\|\boldsymbol{\delta}(k)\|^2 + \mathbb{1}_{(W>T)}\frac{\rho^T-\rho^W}{1-\rho}\zeta\|\boldsymbol{\delta}(T)\|^2}_{\text{Part II}}$$

*where $\rho = 1 - \frac{\alpha}{4L}$, $\zeta = \frac{h^2}{\alpha} + \frac{h^2}{2L}$, and $C_1$ is defined in Theorem 2.*

Next, we discuss the regret bound in Corollary 1, which consists of two parts: Part I involves the variation of the environment $V_T$ and Part II involves the prediction errors $\{\boldsymbol{\delta}(k)\}_{k=1}^{\min(W,T)}$.

**Impact of $V_T$.** With a fixed $V_T$, Part I decays exponentially with the lookahead window $W$. This suggests that the impact of the environment variation $V_T$ on the regret bound decays exponentially as the lookahead window $W$ increases, which is intuitive since long-term thinking/planning allows early preparation for future changes and thus mitigates the negative impact of the environment variation.

**Impact of $\boldsymbol{\delta}(k)$.** Part II in Corollary 1 consists of the prediction error terms in (9) and the prediction error terms in Theorem 1. Notably, for both $W \leq T$ and $W > T$, the factor in front of $\|\boldsymbol{\delta}(k)\|^2$ is dominated by $\rho^{k-1}$ for $1 \leq k \leq \min(W,T)$, which decays exponentially with $k$ since $0 \leq \rho < 1$. This property suggests that the impact of the total $k$-step-ahead prediction error $\|\boldsymbol{\delta}(k)\|^2$ on RHIG's regret bound decays exponentially with $k$, which is intuitive since RHIG (implicitly) focuses more on the shorter-term predictions by using shorter-term predictions in the later iterations of inexact gradient updates. This property also indicates desirable online performance in practice since short-term predictions are usually more accurate and reliable than the long-term ones.

**The order of the regret bound.** Based on the discussions above, the regret bound in Corollary 1 can be summarized as $\tilde{O}(\rho^W \sqrt{V_T T} + \sum_{k=1}^{\min(W,T)} \rho^{k-1} \|\boldsymbol{\delta}(k)\|^2)$. The prediction errors $\|\boldsymbol{\delta}(k)\|^2$ can be either larger or smaller than $V_T$ as mentioned in [23]. When $V_T = o(T)$ and $\|\boldsymbol{\delta}(k)\|^2 = o(T)$ for $k \leq W$, the regret bound of RHIG is sublinear in $T$.[8]

**Choices of $W$.** The optimal choice of $W$ depends on the trade-off between $V_T$ and the prediction errors. For more insightful discussions, we consider non-decreasing $k$-step-ahead prediction errors, i.e. $\|\boldsymbol{\delta}(k)\| \geq \|\boldsymbol{\delta}(k-1)\|$ for $1 \leq k \leq T$ (in practice, longer-term predictions usually suffer worse quality). It can be shown that Part I increases with $V_T$ and Part II increases with the prediction errors. Further, as $W$ increases, Part I decreases but Part II increases.[9] Thus, when Part I dominates the regret bound, i.e. $V_T$ is large when compared with the prediction errors, selecting a large $W$ reduces the regret bound. On the contrary, when Part II dominates the regret bound, i.e. the prediction errors are large when compared with $V_T$, a small $W$ is preferred. The choices of $W$ above are quite intuitive: when the environment is drastically changing while the predictions roughly follow the trends, one should use more predictions to prepare for future changes; however, with poor predictions and slowly changing environments, one can ignore most predictions and rely on the understanding of the current environment. Lastly, though we only consider RHIG with restarted OGD, the discussions provide insights for other $\phi$.

**An upper and a lower bound in a special case.** Next, we consider a special case when $V_T$ is much larger than the prediction errors. It can be shown that the optimal regret is obtained when $W \to +\infty$.

**Corollary 2.** *Consider non-decreasing $k$-step-ahead prediction errors, i.e. $\|\boldsymbol{\delta}(k)\|^2 \geq \|\boldsymbol{\delta}(k-1)\|^2$ for $1 \leq k \leq T$. When $\sqrt{V_T T} \log(1 + \sqrt{T/V_T}) \geq \frac{2Lh^2\rho + \alpha^2\zeta}{2LC_1(1-\rho)\alpha} \|\boldsymbol{\delta}(T)\|^2$, the regret bound is minimized by letting $W \to +\infty$. Further, when $W \to +\infty$, RHIG's regret can be bounded below.*

$$\mathrm{Reg}(RHIG) \leq \frac{\zeta}{1-\rho} \sum_{k=1}^{T} \rho^{k-1} \|\boldsymbol{\delta}(k)\|^2.$$

Since $\sqrt{V_T T} \log(1 + \sqrt{T/V_T})$ increases with $V_T$, the condition in Corollary 2 essentially states that $V_T$ is much larger in comparison to all the prediction errors. Interestingly, the bound in Corollary 2 is not affected by $V_T$, but all prediction errors $\{\|\boldsymbol{\delta}(k)\|^2\}_{k=1}^{T}$ are involved, though the factor of $\|\boldsymbol{\delta}(k)\|^2$ exponentially decays with $k$. Next, we show that such dependence on $\|\boldsymbol{\delta}(k)\|^2$ is unavoidable.

**Theorem 3** (Lower bound for a special case). *For any online algorithm $\mathcal{A}$, there exists nontrivial $\sum_t f(x_t; \theta_t) + d(x_t, x_{t-1})$ and predictions $\theta_{t|t-k}$ satisfying the condition in Corollary 2, with parameters $\rho_0 = (\frac{\sqrt{L} - \sqrt{\alpha}}{\sqrt{L} + \sqrt{\alpha}})^2$, $\zeta_0 = (\frac{h(1-\sqrt{\rho_0})}{\alpha+\beta})^2 \frac{\alpha(1-2\rho_0)}{2} > 0$, such that the regret satisfies:*

$$\mathrm{Reg}(\mathcal{A}) \geq \frac{\zeta_0}{(1-\rho_0)} \sum_{k=1}^{T} \rho_0^{k-1} \|\boldsymbol{\delta}(k)\|^2.$$

In Theorem 3, the influence of $\|\boldsymbol{\delta}(k)\|^2$ also decreases exponentially with $k$, though with a smaller decay factor $\rho_0$. It is left as future work to close the gap between $\rho$ and $\rho_0$ (and between $\zeta$ and $\zeta_0$).

# 5   Stochastic Prediction Errors

In many applications, prediction errors are usually correlated. For example, the predicted market price of tomorrow usually relies on the predicted price of today, which also depends on the price predicted yesterday. Motivated by this, we adopt an insightful and general stochastic model on prediction errors, which was originally proposed in [1]:

$$\delta_t(k) = \theta_t - \theta_{t|t-k} = \sum_{s=t-k+1}^{t} P(t-s)e_s, \quad \forall\, 1 \leq k \leq t \tag{10}$$

where $P(s) \in \mathbb{R}^{p \times q}$, $e_1, \ldots, e_T \in \mathbb{R}^q$ are independent with zero mean and covariance $R_e$. Model (10) captures the correlation patterns described above: the errors $\delta_t(k)$ of different predictions on the same parameter $\theta_t$ are correlated by sharing common random vectors from $\{e_t, \ldots, e_{t-k+1}\}$; and the prediction errors generated at the same stage, i.e. $\theta_{t+k} - \theta_{t+k|t-1}$ for $k \geq 0$, are correlated by sharing common random vectors from $\{e_t, \ldots, e_{t+k}\}$. Notably, the coefficient matrix $P(k)$ represents the degree of correlation between the $\delta_t(1)$ and $\delta_t(k)$ and between $\theta_t - \theta_{t|t-1}$ and $\theta_{t+k} - \theta_{t+k|t-1}$.

As discussed in [1,18], the stochastic model (10) enjoys many applications, e.g. Wiener filters, Kalman filters [36]. For instance, suppose the parameter follows a stochastic linear system: $\theta_t = \gamma\theta_{t-1} + e_t$ with a given $\theta_0$ and random noise $e_t \sim N(0,1)$. Then $\theta_t = \gamma^k\theta_{t-k} + \sum_{s=t-k+1}^{t}\gamma^{t-s}e_s$, the optimal prediction of $\theta_t$ based on $\theta_{t-k}$ is $\theta_{t|t-k} = \gamma^k\theta_{t-k}$, the prediction error $\delta_t(k)$ satisfies the model (10) with $P(t-s) = \gamma^{t-s}$. A large $\gamma$ causes strong correlation among prediction errors.

Our next theorem bounds the expected regret of RHIG by the degree of correlation $\|P(k)\|_F$.

**Theorem 4** (Expected regret bound). *Under Assumption 1-2, $W \geq 0$, $\eta = 1/L$ and initialization $\phi$,*

$$\mathbb{E}[\mathrm{Reg}(RHIG)] \leq \frac{2L}{\alpha}\rho^W \mathbb{E}[\mathrm{Reg}(\phi)] + \sum_{t=0}^{\min(W,T)-1} \zeta\|R_e\|(T-t)\|P(t)\|_F^2 \frac{\rho^t - \rho^W}{1-\rho}$$

*where the expectation is taken with respect to $\{e_t\}_{t=1}^{T}$, $\rho = 1 - \frac{\alpha}{4L}$, $\zeta = \frac{h^2}{\alpha} + \frac{h^2}{2L}$.*

The first term in Theorem 4 represents the influence of $\phi$ while the second term captures the effects of the correlation. We note that the $t$-step correlation $\|P(t)\|_F^2$ decays exponentially with $t$ in the regret bound, indicating that RHIG efficiently handles the strong correlation among prediction errors.

Next, we provide a regret bound when RHIG employs the restarted OGD oracle as in Section 4. Similarly, we consider a known $\mathbb{E}[V_T]$ and $1 \leq V_T \leq T$ for technical simplicity.

**Corollary 3** (RHIG with restarted OGD). *Under Assumption 1-3, consider the restarted OGD with $\Delta = \lceil\sqrt{2T/\mathbb{E}[V_T]}\rceil$, we obtain*

$$\mathbb{E}[\mathrm{Reg}(RHIG)] \leq \rho^W C_2\sqrt{\mathbb{E}[V_T]T}\log(1+\sqrt{T/\mathbb{E}[V_T]}) + \sum_{t=0}^{\min(W,T)-1} \zeta\|R_e\|(T-t)\|P(t)\|_F^2 \frac{\rho^t}{1-\rho},$$

*where we define $C_2 = \frac{2LC_1}{\alpha}$ and $C_1$ is defined in Theorem 2.*

Notice that large $W$ is preferred with a large environment variation and weakly correlated prediction errors, and vice versa.

Next, we discuss the concentration property. For simplicity, we consider Gaussian vectors $\{e_t\}_{t=1}^{T}$.[10]

**Theorem 5** (Concentration bound). *Consider Assumption 1-3 and the conditions in Corollary 3. Let $\mathbb{E}[\mathrm{RegBdd}]$ denote the expected regret bound in Corollary 3 with $\mathbb{E}[V_T] = T$, then we have*

$$\mathbb{P}(\mathrm{Reg}(RHIG) \geq \mathbb{E}[\mathrm{Regbdd}] + b) \leq \exp\left(-c\min\left(\frac{b^2}{K^2}, \frac{b}{K}\right)\right), \quad \forall\, b > 0,$$

*where $K = \zeta\sum_{t=0}^{\min(T,W)-1}\|R_e\|(T-t)\|P(t)\|_F^2 \frac{\rho^t}{1-\rho}$ and $c$ is an absolute constant.*

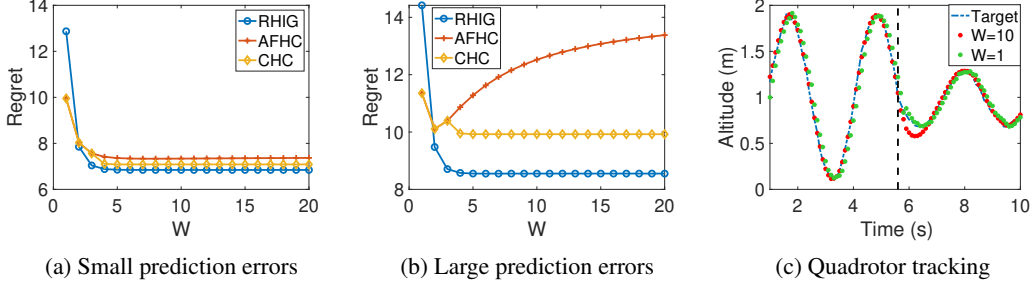

(a) Small prediction errors     (b) Large prediction errors     (c) Quadrotor tracking

Figure 2: (a) and (b): the regrets of RHIG, AFHC and CHC. (c): RHIG's tracking trajectories.

Theorem 5 shows that the probability of the regret being larger than the expected regret by $b > 0$ decays exponentially with $b$ when $\mathbb{E}[V_T] = T$, indicating RHIG's nice concentration property. Further, the concentration effect is stronger (i.e. a larger $1/K$) with a smaller degree of correlation $\|P(t)\|_F^2$.

## 6 Numerical Experiments

We consider online quadrotor tracking of a vertically moving target [37]. We consider (i) a high-level planning problem which is purely online optimization without modeling the physical dynamics; and (ii) a physical tracking problem where simplified quadrotor dynamics are considered [37].

In (i), we consider SOCO: $\min \sum_{t=1}^T \frac{1}{2}(\alpha(x_t - \theta_t)^2 + \beta(x_t - x_{t-1})^2)$, where $x_t$ is quadrotor's altitude, $\theta_t$ is target's altitude, and $(x_t - x_{t-1})^2$ penalizes a sudden change in the quadrotor's altitude. The target $\theta_t$ follows: $\theta_t = y_t + q_t$, where $y_t = \gamma y_{t-1} + e_t$ is an autoregressive process with noise $e_t$ [38] and $q_t = a\sin(\omega t)$ is a periodic signal. The predictions are the sum of $q_t$ and the optimal predictions of $y_t$. Notice that a large $\gamma$ indicates worse long-term predictions. We consider both a small $\gamma = 0.3$ and a large $\gamma = 0.7$ for different levels of errors. We compare RHIG with AFHC [1,6] and CHC [18]. (See the supplementary material for more details.) Figure 2(a) shows that with small prediction errors, the three algorithms perform similarly well and RHIG is slightly better. Figure 2(b) shows that with large prediction errors, RHIG significantly outperforms AFHC and CHC. Some intuitive explanations are provided below. Firstly, AFHC and CHC are optimization-based methods, while our RHIG is based on gradient descent, which is known to be more robust to errors. Secondly, RHIG implicitly reduces the impact of the (poorer-quality) long-term predictions and focuses more on the (better) short-term ones by using long-term predictions in the first several updates and then using short-term ones in later updates to refine the decisions; while AFHC and CHC treat predictions more equally by taking averages of the optimal solutions computed by both long-term and short-term predictions (see [1,18] for more details). These two intuitive reasons may explain the better numerical performance of our RHIG when compared with AFHC and CHC.

In (ii), we consider a simplified second-order model of quadrotor vertical flight: $\ddot{x} = k_1 u - g + k_2$, where $x, \dot{x}, \ddot{x}$ are the altitude, velocity and acceleration respectively, $u$ is the control input (motor thrust command), $g$ is the gravitational acceleration, $k_1$ and $k_2$ are physical parameters. We consider time discretization and cost function $\sum_{t=1}^T \frac{1}{2}(\alpha(x_t - \theta_t)^2 + \beta u_t^2)$. The target $\theta_t$ follows the process in (i), but with a sudden change in $q_t$ at $t_c = 5.6$s, causing large prediction errors at around $t_c$, which is unknown until $t_c$. Figure 2(c) plots the quadrotor's trajectories generated by RHIG with $W = 1, 10$ and shows RHIG's nice tracking performance even when considering physical dynamics. $W = 10$ performs better first by using more predictions. However, right after $t_c$, $W = 1$ performs better since the poor prediction quality there degrades the performance. Lastly, the trajectory with $W = 10$ quickly returns to the desired one after $t_c$, showing the robustness of RHIG to prediction error shocks.

## 7 Conclusion

This paper studies how to leverage multi-step-ahead noisy predictions in smoothed online convex optimization. We design a gradient-based algorithm RHIG and analyze its dynamic regret under general prediction errors and a stochastic prediction error model. RHIG effectively reduces the impact of multi-step-ahead prediction errors. Future work includes: 1) closing the gap between the upper and the lower bound in Section 4; 2) lower bounds for general cases; 3) online control problems; 4) the convex case analysis without the strong convexity assumption; etc.

## Broader Impact

This paper conducts foundational research and theoretical study on online (real-time) decision making problems. In particular, we propose an online algorithm that leverages noisy predictions in online decision making with coupling among stages. This work can be potentially applied to real-time planning problems in e.g. data center management, robotics, smart grids, smart buildings, transportation systems, as well as other online control applications. Though the algorithm is promising and the analysis is insightful, the results are limited by the theoretical assumptions and should be carefully tested and adjusted before being used in real systems. Further, this paper focuses on the efficiency (i.e. the dynamic regret) and does not consider societal issues such as fairness and privacy. Lastly, we see no ethical concerns on this paper.

## Acknowledgments and Disclosure of Funding

The work was supported by NSF CAREER 1553407, AFOSR YIP, ONR YIP.

## Footnotes

[1]If only $W$-step-ahead predictions are received, we define $\theta_{t+\tau|t-1} := \theta_{t+W-1|t-1}$ for $\tau \geq W$.

[2]For instance, if OGD is used as the initialization oracle $\phi$, then $x_{t+W}(0) = x_{t+W-1}(0) - \xi_{t+W} \nabla_x f(x_{t+W-1}(0); \theta_{t+W-1})$, where $\xi_{t+W}$ denotes the stepsize.

[3]For more discussion on $W > T$, we refer the reader to our supplementary material.

[4]The results in this section can be extended to more general time-varying cost functions, i.e. $f_t(\cdot)$, where the prediction errors will be measured by the difference in the gradients, i.e. $\sup_{x \in \mathbb{X}} \|\nabla f_t(x) - \nabla f_{t\mid t-k}(x)\|$.

[5]This is without loss of generality. When $V_T$ is unknown, we can use doubling tricks and adaptive stepsizes to generate similar bounds [23]. $1 \leq V_T \leq T$ can be enforced by defining a proper $\theta_0$ and by normalization.

[6]Other norms work too, only leading to different constant factors in the regret bounds.

[7]We have this error term because we do not impose the stochastic structures of the gradient errors in [34].

[8] For example, consider $\theta_{t-1}$ as the prediction of $\theta_{t+k}$ at time $t$ for $k \geq 0$, then $\|\boldsymbol{\delta}(k)\|^2 = O(\sum_{t=1}^{T} \|\theta_t - \theta_{t-k-1}\|^2)$. If $\Theta$ is bounded and $\sum_{t=1}^{T} \|\theta_t - \theta_{t-1}\| = o(T)$, then $\|\boldsymbol{\delta}(k)\|^2 = o(T)$. Further, if $f(x; \theta)$ is Lipschitz continuous with respect to $\theta$, then $V_T = o(T)$. In this case, the regret of RHIG is $o(T)$.

[9] All the monotonicity claims above are verified in the supplementary file and omitted here for brevity.

[10]Similar results can be obtained for sub-Gaussian random vectors.

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
