[Supplementary Material]

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

[11]Notice that $y_t^*$ is a random variable depending on $\theta_t$ for all $t$. Therefore, in the inequalities below, $\mathbb{E}[f(y_{k\Delta+1}^*; \theta_t) - f(y_{k\Delta+1}^*; \theta_{k\Delta+1})]$ can be larger than the term $\sup_{x \in \mathbb{X}} \mathbb{E}[f(x; \theta_t) - f(x; \theta_{k\Delta+1})]$, where $x$ is restricted to only deterministic variables. Thus, the expectation operator $\mathbb{E}$ must be outside the $\sup$ operator in our definition of the expected variation of the environment (see $\mathbb{E}[V^k]$ and $\mathbb{E}[V_T]$).

[12]Here we use the fact that $\|X_i\|_\varphi = 1$ where $\|\cdot\|_\varphi$ is the subGaussian norm defined in [39].

[13]An absolute constant refers to a quantity that does not change with anything.

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

## Appendices

The appendices provide additional discussions and proofs of the theoretical results. In particular, Appendix A provides additional discussions on RHIG when $W > T$; next, Appendix B provides a proof of Lemma 1; Appendix C provides a proof of Theorem 1; Appendix D provides a proof of Theorem 2, a proof of Corollary 1, and also proves the claimed properties of the regret bound in Section 4; Appendix E discusses the special case and proves Corollary 2 and Theorem 3; Appendix F considers the stochastic prediction errors and proves Theorem 4, Corollary 3 and Theorem 5; finally, Appendix G provides additional discussions on the numerical experiments.

## A  Additional Discussions on RHIG when $W > T$

As mentioned in Section 3, in RHIG, one can select the lookahead horizon $W > T$. To further illustrate this case, we provide an example of RHIG for $T = 3$ and $W = 5$ in Figure 3.

Further, we explain the case for general $W > T$ and when $W \to +\infty$ by using the example in Figure 3 for $T = 3$. Similar to the discussion for Figure 3, for general $W > T$, it can be verified that the computed variables at $t = 0$ are $x_1(W - 1), x_2(W - 2), x_3(W - 3)$, which can be viewed as the iterated variables of offline (exact) gradient descent (4) under parameters $(\theta_{1|0}, \theta_{2|0}, \theta_{3|0})$ after $W - 1, W - 2, W - 3$ iterations respectively. Then, at $t = 1$, RHIG conducts one inexact gradient update to obtain $x_3(W - 2), x_2(W - 1), x_1(W)$ based on predictions $(\theta_{1|0}, \theta_{2|0}, \theta_{3|0})$ and outputs $x_1(W)$. At $t = 2$, RHIG conducts one inexact gradient update to obtain $x_3(W - 1)$ and $x_2(W)$ based on predictions $\theta_{3|1}$ and $\theta_{2|2}$. At $t = 3$, RHIG conducts one exact gradient update to obtain $x_3(W)$ based on prediction $\theta_{3|2}$. When $W \to +\infty$, $(x_1(W - 1), x_2(W - 2), x_3(W - 3))$ converges to the optimal solution to $\min_{\bm{x} \in \mathbb{X}^T} C(\bm{x}; (\theta_{1|0}, \theta_{2|0}, \theta_{3|0}))$. Then, at $t = 1$, $(x_3(W - 2), x_2(W - 1), x_1(W))$ is the same as $(x_1(W - 1), x_2(W - 2), x_3(W - 3))$ since it has converged to $\min_{\bm{x} \in \mathbb{X}^T} C(\bm{x}; (\theta_{1|0}, \theta_{2|0}, \theta_{3|0}))$. Nevertheless, at $t = 2$, RHIG will conduct one inexact gradient update to obtain new $x_3(W - 1)$ and $x_2(W)$ because there are *new* predictions $\theta_{3|1}$ and $\theta_{2|2}$ available. Similarly, at $t = 3$, RHIG conducts one exact gradient update to obtain a new $x_3(W)$ based on *new* prediction $\theta_{3|2}$.

## B  Proof of Lemma 1

Firstly, we prove the strong convexity. Since $f(x_t; \theta_t)$ is $\alpha$-strongly convex with respect to $x_t$, $\sum_{t=1}^{T} f(x_t; \theta_t)$ is $\alpha$-strongly convex with respect to $\bm{x} = (x_1^\top, \dots, x_T^\top)^\top$. Since $d(x_t, x_{t-1})$ is convex with respect to $(x_t, x_{t-1})$, $\sum_{t=1}^{T} d(x_t, x_{t-1})$ is also convex with respect to $\bm{x}$. Consequently, $C(\bm{x}; \bm{\theta}) = \sum_{t=1}^{T} (f(x_t; \theta_t) + d(x_t, x_{t-1}))$ is $\alpha$-strongly convex with respect to $\bm{x}$.

| $x_1(0) = x_0$ | $x_2(0); \phi$ | $x_3(0); \phi$ |
| $x_1(1); \theta_{1|-4}$ | $x_2(1); \theta_{2|-3}$ | $x_3(1); \theta_{3|-2}$ |
| $x_1(2); \theta_{1|-3}$ | $x_2(2); \theta_{2|-2}$ | $x_3(2); \theta_{3|-1}$ |
| $x_1(3); \theta_{1|-2}$ | $x_2(3); \theta_{2|-1}$ | $x_3(3); \theta_{3|0}$ |
| $x_1(4); \theta_{1|-1}$ | $x_2(4); \theta_{2|0}$ | $x_3(4); \theta_{3|1}$ |
| $x_1(5); \theta_{1|0}$ | $x_2(5); \theta_{2|1}$ | $x_3(5); \theta_{3|2}$ |

Figure 3: Example of RHIG when $W = 5 > T = 3$. (Pink) At $t = 1 - W = -4$, let $x_1(0) = x_0$. (Orange) At $t = -3$, initialize $x_2(0)$ by $\phi$, then compute $x_1(1)$ by inexact offline GD (5) with prediction $\theta_{1|t-1} = \theta_{1|-4} = \theta_{1|0}$. (Yellow) At $t = -2$, initialize $x_3(0)$ by $\phi$, and update $x_2(1)$ and $x_1(2)$ by (5) with $\theta_{2|-3} = \theta_{2|0}$ and $\theta_{1|-3} = \theta_{1|0}$ respectively. (Green) At $t = -1$, update $x_3(1)$, $x_2(2), x_1(3)$ by inexact offline GD (5) with $\theta_{3|-2} = \theta_{3|0}, \theta_{2|-2} = \theta_{2|0}$, and $\theta_{1|-2} = \theta_{1|0}$ respectively. (Dark green) At $t = 0$, update $x_3(2), x_2(3), x_1(4)$ by inexact offline GD (5) with $\theta_{3|-1} = \theta_{3|0}$, $\theta_{2|-1} = \theta_{2|0}$, and $\theta_{1|12} = \theta_{1|0}$ respectively. (Blue) At $t = 1$, update $x_3(3), x_2(4), x_1(5)$ by inexact offline GD (5) with $\theta_{3|0}, \theta_{2|0}$, and $\theta_{1|0}$ respectively. Then, RHIG outputs $x_1(5)$. (Purple) At $t = 2$, update $x_3(4)$ and $x_2(5)$ by inexact offline GD (5) with $\theta_{3|1}, \theta_{2|1}$ respectively and output $x_2(5)$. (Red) At $t = 3$, update $x_3(5)$ by inexact offline GD (5) with $\theta_{3|2}$ and output $x_3(5)$.

By recalling that $\theta_{t|\tau} = \theta_{t|0}$ when $\tau < 0$, we note that all the computation at $t \leq 0$ (above the red lines) is based on initial predictions $\{\theta_{1|0}, \theta_{2|0}, \theta_{3|0}\}$. Therefore, the computed variables $x_1(4), x_2(3), x_3(2)$ at $t = 0$ can be viewed as the iterated variables of offline (exact) gradient descent (4) under parameters $\{\theta_{1|0}, \theta_{2|0}, \theta_{3|0}\}$ after 4,3,2 iterations respectively.

Next, we prove the smoothness. For any $x_t, y_t \in \mathbb{X}$, by the $l_f$-smoothness of $f(x_t; \theta_t)$ for all $t$, we have

$$f(y_t) \leq f(x_t) + \langle \nabla_{x_t} f(x_t; \theta_t), y_t - x_t \rangle + \frac{l_f}{2}\|x_t - y_t\|^2$$

By the $l_d$-smoothness of $d(x_t, x_{t-1})$, for any $x_t, y_t, x_{t-1}, y_{t-1} \in \mathbb{X}$, we have

$$d(y_t, y_{t-1}) \leq d(x_t, x_{t-1}) + \langle \nabla_{x_t} d(x_t, x_{t-1}), y_t - x_t \rangle + \langle \nabla_{x_{t-1}} d(x_t, x_{t-1}), y_{t-1} - x_{t-1} \rangle$$
$$+ \frac{l_d}{2}(\|y_t - x_t\|^2 + \|y_{t-1} - x_{t-1}\|^2)$$

for $t \geq 2$ and $d(y_1, x_0) \leq d(x_1, x_0) + \langle \nabla_{x_1} d(x_1, x_0), y_1 - x_1 \rangle + \frac{l_d}{2}\|x_1 - y_1\|^2$ for $t = 1$.

Therefore, for any $\boldsymbol{x}, \boldsymbol{y} \in \mathbb{X}^T$, by summing the smoothness inequalities above over $t = 1, \ldots, T$, we obtain

$$C(\boldsymbol{y}; \boldsymbol{\theta}) \leq C(\boldsymbol{x}; \boldsymbol{\theta}) + \langle \nabla_{\boldsymbol{x}} C(\boldsymbol{x}; \boldsymbol{\theta}), \boldsymbol{y} - \boldsymbol{x} \rangle + \frac{l_f + 2l_d}{2}\|\boldsymbol{x} - \boldsymbol{y}\|^2$$

where we used the fact that $\nabla_{\boldsymbol{x}} C(\boldsymbol{x}; \boldsymbol{\theta})$ is composed of partial gradients $\nabla_{x_t} C(\boldsymbol{x}; \boldsymbol{\theta}) = \nabla_{x_t} f(x_t; \theta_t) + \nabla_{x_t} d(x_t, x_{t-1}) + \mathbb{1}_{(t<T)} \cdot \nabla_{x_t} d(x_{t+1}, x_t)$.

## C Proof of Theorem 1

Consider the offline optimization with parameter $\boldsymbol{\theta}$, i.e. $\min_{\boldsymbol{x} \in \mathbb{X}^T} C(\boldsymbol{x}; \boldsymbol{\theta})$. As mentioned in Section 3, RHIG can be interpreted as projected gradient descent on $C(\boldsymbol{x}; \boldsymbol{\theta})$ with inexact gradients:

$$\boldsymbol{x}(k+1) = \Pi_{\mathbb{X}^T} \left[ \boldsymbol{x}(k) - \eta \nabla_{\boldsymbol{x}} C(\boldsymbol{x}(k); \boldsymbol{\theta} - \boldsymbol{\delta}(W-k)) \right] \tag{11}$$

where the exact gradient should be $\nabla_{\boldsymbol{x}} C(\boldsymbol{x}(k); \boldsymbol{\theta})$ but the parameter prediction error $\boldsymbol{\delta}(W-k)$ results in inexact gradient $\nabla_{\boldsymbol{x}} C(\boldsymbol{x}(k); \boldsymbol{\theta} - \boldsymbol{\delta}(W-k))$. Notice that when $W - k > T$, by our definition, we have $\boldsymbol{\delta}(W-k) = \boldsymbol{\delta}(T)$.

Consequently, the regret bound of RHIG can be proved based on the convergence analysis of the projected gradient descent with inexact gradients. We note that unlike the classic inexact gradient

where the gradient errors are uniformly bounded, RHIG's inexact gradients (11) have different gradient errors at different iterations, thus calling for slightly different convergence analysis.

In the following, we first provide some supportive lemmas, then provide a rigorous proof of Theorem 1.

## C.1 Supportive Lemmas

Firstly, we provide a bound on the gradient errors with respect to the errors on the parameters.

**Lemma 2** (Gradient prediction error bound). *For any true parameter $\boldsymbol{\theta} \in \Theta^T$ and the predicted parameter $\boldsymbol{\theta}' \in \Theta^T$, the error of the predicted gradient can be bounded below.*

$$\|\nabla_{\boldsymbol{x}} C(\boldsymbol{x}; \boldsymbol{\theta}') - \nabla_{\boldsymbol{x}} C(\boldsymbol{x}; \boldsymbol{\theta})\|^2 \leq h^2 \|\boldsymbol{\theta}' - \boldsymbol{\theta}\|^2, \quad \forall \, \boldsymbol{x} \in \Theta^T$$

*Proof.* Firstly, we consider the gradient with respect to each stage variable $x_t$, which is provided by

$$\nabla_{x_t} C(\boldsymbol{x}; \boldsymbol{\theta}) = \nabla_{x_t} f(x_t; \theta_t) + \nabla_{x_t} d(x_t, x_{t-1}) + \nabla_{x_t} d(x_{t+1}, x_t) \mathbb{1}_{(t \leq T-1)}$$

Noticing that $d(x_t, x_{t-1})$ does not depend on the parameter $\boldsymbol{\theta}$, we obtain the prediction error bound of gradient with respect to $x_t$ as follows.

$$\begin{aligned}\|\nabla_{x_t} C(\boldsymbol{x}; \boldsymbol{\theta}') - \nabla_{x_t} C(\boldsymbol{x}; \boldsymbol{\theta})\| &= \|\nabla_{x_t} f(x_t; \theta_t') - \nabla_{x_t} f(x_t; \theta_t)\| \\ &\leq h\|\theta_t' - \theta_t\|\end{aligned}$$

Therefore, the prediction error of the full gradient can be bounded as follows,

$$\begin{aligned}\|\nabla_{\boldsymbol{x}} C(\boldsymbol{x}; \boldsymbol{\theta}') - \nabla_{\boldsymbol{x}} C(\boldsymbol{x}; \boldsymbol{\theta})\|^2 &= \sum_{t=1}^{T} \|\nabla_{x_t} C(\boldsymbol{x}; \boldsymbol{\theta}') - \nabla_{x_t} C(\boldsymbol{x}; \boldsymbol{\theta})\|^2 \\ &\leq h^2 \sum_{t=1}^{T} \|\theta_t' - \theta_t\|^2 = h^2 \|\boldsymbol{\theta}' - \boldsymbol{\theta}\|^2\end{aligned}$$

which completes the proof. $\square$

Next, we provide an equivalent characterization of the projected gradient update with respect to inexact parameters.

**Lemma 3** (A representation of inexact projected gradient updates). *For any predicted parameter $\boldsymbol{\theta}'$ and any stepsize $\eta$, the projected gradient descent with predicted parameter $\boldsymbol{x}(k+1) = \Pi_{\mathbb{X}^T} [\boldsymbol{x}(k) - \eta \nabla_{\boldsymbol{x}} C(\boldsymbol{x}(k); \boldsymbol{\theta}')]$ is equivalent to the following representation.*

$$\boldsymbol{x}(k+1) = \arg\min_{\boldsymbol{x} \in \mathbb{X}^T} \left\{ \langle \nabla_{\boldsymbol{x}} C(\boldsymbol{x}(k); \boldsymbol{\theta}'), \boldsymbol{x} - \boldsymbol{x}(k) \rangle + \frac{1}{2\eta} \|\boldsymbol{x} - \boldsymbol{x}(k)\|^2 \right\}$$

*Proof.* By the definition of projection, the projected gradient descent with predicted parameter is equivalent to the following.

$$\begin{aligned}\boldsymbol{x}(k+1) &= \arg\min_{\boldsymbol{x} \in \mathbb{X}^T} \left\{ \|\boldsymbol{x} - \boldsymbol{x}(k) + \eta \nabla_{\boldsymbol{x}} C(\boldsymbol{x}(k); \boldsymbol{\theta}')\|^2 \right\} \\ &= \arg\min_{\boldsymbol{x} \in \mathbb{X}^T} \left\{ \|\boldsymbol{x} - \boldsymbol{x}(k)\|^2 + \eta^2 \|\nabla_{\boldsymbol{x}} C(\boldsymbol{x}(k); \boldsymbol{\theta}')\|^2 + 2\eta \langle \nabla_{\boldsymbol{x}} C(\boldsymbol{x}(k); \boldsymbol{\theta}'), \boldsymbol{x} - \boldsymbol{x}(k) \rangle \right\} \\ &= \arg\min_{\boldsymbol{x} \in \mathbb{X}^T} \left\{ \frac{1}{2\eta} \|\boldsymbol{x} - \boldsymbol{x}(k)\|^2 + \langle \nabla_{\boldsymbol{x}} C(\boldsymbol{x}(k); \boldsymbol{\theta}'), \boldsymbol{x} - \boldsymbol{x}(k) \rangle \right\}\end{aligned}$$

where the last equality uses the fact that $\eta^2 \|\nabla_{\boldsymbol{x}} C(\boldsymbol{x}(k); \boldsymbol{\theta}')\|^2$ does not depend on $\boldsymbol{x}$. $\square$

Lastly, we provide a strong-convexity-type inequality and a smoothness-type inequality under inexact gradients. Both inequalities suffer from additional error terms caused by the parameter prediction error.

**Lemma 4** (Strong convexity inequality with errors). *Consider optimization $\min_{\boldsymbol{x}\in\mathbb{X}^T} C(\boldsymbol{x};\boldsymbol{\theta})$. For any $\boldsymbol{x},\boldsymbol{y}\in\mathbb{X}^T$, for any inexact parameter $\boldsymbol{\theta}'$ and the resulting inexact gradient $\nabla_{\boldsymbol{x}}C(\boldsymbol{x};\boldsymbol{\theta}')$, we have*

$$C(\boldsymbol{y};\boldsymbol{\theta}) \geq C(\boldsymbol{x};\boldsymbol{\theta}) + \langle \nabla_{\boldsymbol{x}}C(\boldsymbol{x};\boldsymbol{\theta}'), \boldsymbol{y}-\boldsymbol{x}\rangle + \frac{\alpha}{4}\|\boldsymbol{x}-\boldsymbol{y}\|^2 - \frac{h^2}{\alpha}\|\boldsymbol{\theta}'-\boldsymbol{\theta}\|^2$$

*Proof.* By the strong convexity of $C(\boldsymbol{x};\boldsymbol{\theta})$, for any $\boldsymbol{x},\boldsymbol{y}\in\mathbb{X}^T$ and any $\boldsymbol{\theta},\boldsymbol{\theta}'\in\Theta^T$, we obtain the following.

$$
\begin{aligned}
C(\boldsymbol{y};\boldsymbol{\theta}) &\geq C(\boldsymbol{x};\boldsymbol{\theta}) + \langle \nabla_{\boldsymbol{x}}C(\boldsymbol{x};\boldsymbol{\theta}), \boldsymbol{y}-\boldsymbol{x}\rangle + \frac{\alpha}{2}\|\boldsymbol{y}-\boldsymbol{x}\|^2 \\
&= C(\boldsymbol{x};\boldsymbol{\theta}) + \langle \nabla_{\boldsymbol{x}}C(\boldsymbol{x};\boldsymbol{\theta}'), \boldsymbol{y}-\boldsymbol{x}\rangle - \langle \nabla_{\boldsymbol{x}}C(\boldsymbol{x};\boldsymbol{\theta}') - \nabla_{\boldsymbol{x}}C(\boldsymbol{x};\boldsymbol{\theta}), \boldsymbol{y}-\boldsymbol{x}\rangle + \frac{\alpha}{2}\|\boldsymbol{y}-\boldsymbol{x}\|^2 \\
&\geq C(\boldsymbol{x};\boldsymbol{\theta}) + \langle \nabla_{\boldsymbol{x}}C(\boldsymbol{x};\boldsymbol{\theta}'), \boldsymbol{y}-\boldsymbol{x}\rangle - \|\nabla_{\boldsymbol{x}}C(\boldsymbol{x};\boldsymbol{\theta}') - \nabla_{\boldsymbol{x}}C(\boldsymbol{x};\boldsymbol{\theta})\|\|\boldsymbol{y}-\boldsymbol{x}\| + \frac{\alpha}{2}\|\boldsymbol{y}-\boldsymbol{x}\|^2 \\
&\geq C(\boldsymbol{x};\boldsymbol{\theta}) + \langle \nabla_{\boldsymbol{x}}C(\boldsymbol{x};\boldsymbol{\theta}'), \boldsymbol{y}-\boldsymbol{x}\rangle - \frac{1}{\alpha}\|\nabla_{\boldsymbol{x}}C(\boldsymbol{x};\boldsymbol{\theta}') - \nabla_{\boldsymbol{x}}C(\boldsymbol{x};\boldsymbol{\theta})\|^2 + \frac{\alpha}{4}\|\boldsymbol{y}-\boldsymbol{x}\|^2 \\
&\geq C(\boldsymbol{x};\boldsymbol{\theta}) + \langle \nabla_{\boldsymbol{x}}C(\boldsymbol{x};\boldsymbol{\theta}'), \boldsymbol{y}-\boldsymbol{x}\rangle - \frac{h^2}{\alpha}\|\boldsymbol{\theta}'-\boldsymbol{\theta}\|^2 + \frac{\alpha}{4}\|\boldsymbol{y}-\boldsymbol{x}\|^2
\end{aligned}
$$

$\square$

**Lemma 5** (Smoothness inequality with errors). *Consider optimization $\min_{\boldsymbol{x}\in\mathbb{X}^T} C(\boldsymbol{x};\boldsymbol{\theta})$. For any $\boldsymbol{x},\boldsymbol{y}\in\mathbb{X}^T$, for any inexact parameter $\boldsymbol{\theta}'$ and the resulting inexact gradient $\nabla_{\boldsymbol{x}}C(\boldsymbol{x};\boldsymbol{\theta}')$, we have*

$$C(\boldsymbol{y};\boldsymbol{\theta}) \leq C(\boldsymbol{x};\boldsymbol{\theta}) + \langle \nabla_{\boldsymbol{x}}C(\boldsymbol{x};\boldsymbol{\theta}'), \boldsymbol{y}-\boldsymbol{x}\rangle + L\|\boldsymbol{x}-\boldsymbol{y}\|^2 + \frac{h^2}{2L}\|\boldsymbol{\theta}'-\boldsymbol{\theta}\|^2$$

*Proof.* By the smoothness of $C(\boldsymbol{x};\boldsymbol{\theta})$, for any $\boldsymbol{x},\boldsymbol{y}\in\mathbb{X}^T$ and any $\boldsymbol{\theta},\boldsymbol{\theta}'\in\Theta^T$, we obtain the following.

$$
\begin{aligned}
C(\boldsymbol{y};\boldsymbol{\theta}) &\leq C(\boldsymbol{x};\boldsymbol{\theta}) + \langle \nabla_{\boldsymbol{x}}C(\boldsymbol{x};\boldsymbol{\theta}), \boldsymbol{y}-\boldsymbol{x}\rangle + \frac{L}{2}\|\boldsymbol{y}-\boldsymbol{x}\|^2 \\
&= C(\boldsymbol{x};\boldsymbol{\theta}) + \langle \nabla_{\boldsymbol{x}}C(\boldsymbol{x};\boldsymbol{\theta}'), \boldsymbol{y}-\boldsymbol{x}\rangle + \langle \nabla_{\boldsymbol{x}}C(\boldsymbol{x};\boldsymbol{\theta}) - \nabla_{\boldsymbol{x}}C(\boldsymbol{x};\boldsymbol{\theta}'), \boldsymbol{y}-\boldsymbol{x}\rangle + \frac{L}{2}\|\boldsymbol{y}-\boldsymbol{x}\|^2 \\
&\leq C(\boldsymbol{x};\boldsymbol{\theta}) + \langle \nabla_{\boldsymbol{x}}C(\boldsymbol{x};\boldsymbol{\theta}'), \boldsymbol{y}-\boldsymbol{x}\rangle + \|\nabla_{\boldsymbol{x}}C(\boldsymbol{x};\boldsymbol{\theta}') - \nabla_{\boldsymbol{x}}C(\boldsymbol{x};\boldsymbol{\theta})\|\|\boldsymbol{y}-\boldsymbol{x}\| + \frac{L}{2}\|\boldsymbol{y}-\boldsymbol{x}\|^2 \\
&\leq C(\boldsymbol{x};\boldsymbol{\theta}) + \langle \nabla_{\boldsymbol{x}}C(\boldsymbol{x};\boldsymbol{\theta}'), \boldsymbol{y}-\boldsymbol{x}\rangle + \frac{1}{2L}\|\nabla_{\boldsymbol{x}}C(\boldsymbol{x};\boldsymbol{\theta}') - \nabla_{\boldsymbol{x}}C(\boldsymbol{x};\boldsymbol{\theta})\|^2 + L\|\boldsymbol{y}-\boldsymbol{x}\|^2 \\
&\leq C(\boldsymbol{x};\boldsymbol{\theta}) + \langle \nabla_{\boldsymbol{x}}C(\boldsymbol{x};\boldsymbol{\theta}'), \boldsymbol{y}-\boldsymbol{x}\rangle + \frac{h^2}{2L}\|\boldsymbol{\theta}'-\boldsymbol{\theta}\|^2 + L\|\boldsymbol{y}-\boldsymbol{x}\|^2
\end{aligned}
$$

$\square$

## C.2    Proof of Theorem 1

According to Algorithm 2 and the definition of the regret, we have $\text{Reg}(RHIG) = C(\boldsymbol{x}(W);\boldsymbol{\theta}) - C(\boldsymbol{x}^*;\boldsymbol{\theta})$ and $\text{Reg}(\phi) = C(\boldsymbol{x}(0);\boldsymbol{\theta}) - C(\boldsymbol{x}^*;\boldsymbol{\theta})$, where $\boldsymbol{x}^* = \arg\min_{\mathbb{X}^T} C(\boldsymbol{x};\boldsymbol{\theta})$. For notational simplicity, we denote $r_k = \|\boldsymbol{x}(k) - \boldsymbol{x}^*\|^2$.

*Step 1: bound $\text{Reg}(RHIG)$ with $r_{W-1}$.*

$$
\begin{aligned}
C(\boldsymbol{x}(W);\boldsymbol{\theta}) &\leq C(\boldsymbol{x}(W-1);\boldsymbol{\theta}) + \langle \nabla_{\boldsymbol{x}}C(\boldsymbol{x}(W-1);\boldsymbol{\theta}-\boldsymbol{\delta}(1)), \boldsymbol{x}(W)-\boldsymbol{x}(W-1)\rangle \\
&\quad + L\|\boldsymbol{x}(W)-\boldsymbol{x}(W-1)\|^2 + \frac{h^2}{2L}\|\boldsymbol{\delta}(1)\|^2 \\
&= \min_{\boldsymbol{x}\in\mathbb{X}^T}\left\{\langle \nabla_{\boldsymbol{x}}C(\boldsymbol{x}(W-1);\boldsymbol{\theta}-\boldsymbol{\delta}(1)), \boldsymbol{x}-\boldsymbol{x}(W-1)\rangle + L\|\boldsymbol{x}-\boldsymbol{x}(W-1)\|^2\right\} \\
&\quad + C(\boldsymbol{x}(W-1);\boldsymbol{\theta}) + \frac{h^2}{2L}\|\boldsymbol{\delta}(1)\|^2 \\
&\leq \langle \nabla_{\boldsymbol{x}}C(\boldsymbol{x}(W-1);\boldsymbol{\theta}-\boldsymbol{\delta}(1)), \boldsymbol{x}^*-\boldsymbol{x}(W-1)\rangle + L\|\boldsymbol{x}^*-\boldsymbol{x}(W-1)\|^2
\end{aligned}
$$

$$+ C(\boldsymbol{x}(W-1); \boldsymbol{\theta}) + \frac{h^2}{2L}\|\boldsymbol{\delta}(1)\|^2$$

$$\leq C(\boldsymbol{x}^*; \boldsymbol{\theta}) + (L - \frac{\alpha}{4})r_{W-1} + \left(\frac{h^2}{\alpha} + \frac{h^2}{2L}\right)\|\boldsymbol{\delta}(1)\|^2$$

where we used Lemma 5 in the first inequality, Lemma 3 and $\eta = \frac{1}{2L}$ in the first equality, Lemma 4 in the last inequality. By rearranging terms, we obtain

$$\text{Reg}(RHIG) = C(\boldsymbol{x}(W); \boldsymbol{\theta}) - C(\boldsymbol{x}^*; \boldsymbol{\theta}) \leq L\rho r_{W-1} + \zeta\|\boldsymbol{\delta}(1)\|^2 \tag{12}$$

where $\rho = 1 - \frac{\alpha}{4L}$, $\zeta = \frac{h^2}{\alpha} + \frac{h^2}{2L}$.

*Step 2: a recursive inequality between $r_{k+1}$ and $r_k$.*
In the following, we will show that

$$r_{k+1} \leq \rho r_k + \frac{\zeta}{L}\|\boldsymbol{\delta}(W-k)\|^2, \quad \forall\, 0 \leq k \leq W-1 \tag{13}$$

Firstly, by (11), $\eta = \frac{1}{2L}$, Lemma 3 and its first-order optimality condition, we have

$$\langle \nabla_{\boldsymbol{x}} C(\boldsymbol{x}(k); \boldsymbol{\theta} - \delta(W-k)) + 2L(\boldsymbol{x}(k+1) - \boldsymbol{x}(k)), \boldsymbol{x} - \boldsymbol{x}(k+1) \rangle \geq 0, \quad \forall\, \boldsymbol{x} \in \mathbb{X}^T$$

By substituting $\boldsymbol{x} = \boldsymbol{x}^*$ and rearranging terms, we obtain

$$\frac{1}{2L}\langle \nabla_{\boldsymbol{x}} C(\boldsymbol{x}(k); \boldsymbol{\theta} - \delta(W-k)), \boldsymbol{x}^* - \boldsymbol{x}(k+1) \rangle \geq \langle \boldsymbol{x}(k+1) - \boldsymbol{x}(k), \boldsymbol{x}(k+1) - \boldsymbol{x}^* \rangle \tag{14}$$

Next, we will derive the recursive inequality (13) by using (14).

$$r_{k+1} = \|\boldsymbol{x}(k+1) - \boldsymbol{x}^*\|^2 = \|\boldsymbol{x}(k+1) - \boldsymbol{x}(k) + \boldsymbol{x}(k) - \boldsymbol{x}^*\|^2$$

$$= r_k - \|\boldsymbol{x}(k+1) - \boldsymbol{x}(k)\|^2 + 2\langle \boldsymbol{x}(k+1) - \boldsymbol{x}(k), \boldsymbol{x}(k+1) - \boldsymbol{x}^* \rangle$$

$$\leq r_k - \|\boldsymbol{x}(k+1) - \boldsymbol{x}(k)\|^2 + \frac{1}{L}\langle \nabla_{\boldsymbol{x}} C(\boldsymbol{x}(k); \boldsymbol{\theta} - \boldsymbol{\delta}(W-k)), \boldsymbol{x}^* - \boldsymbol{x}(k+1) \rangle$$

$$= r_k - \|\boldsymbol{x}(k+1) - \boldsymbol{x}(k)\|^2 + \frac{1}{L}\langle \nabla_{\boldsymbol{x}} C(\boldsymbol{x}(k); \boldsymbol{\theta} - \boldsymbol{\delta}(W-k)), \boldsymbol{x}^* - \boldsymbol{x}(k) \rangle$$

$$\quad + \frac{1}{L}\langle \nabla_{\boldsymbol{x}} C(\boldsymbol{x}(k); \boldsymbol{\theta} - \boldsymbol{\delta}(W-k)), \boldsymbol{x}(k) - \boldsymbol{x}(k+1) \rangle$$

$$= r_k + \frac{1}{L}\langle \nabla_{\boldsymbol{x}} C(\boldsymbol{x}(k); \boldsymbol{\theta} - \boldsymbol{\delta}(W-k)), \boldsymbol{x}^* - \boldsymbol{x}(k) \rangle$$

$$\quad - \frac{1}{L}\left(\langle \nabla_{\boldsymbol{x}} C(\boldsymbol{x}(k); \boldsymbol{\theta} - \boldsymbol{\delta}(W-k)), \boldsymbol{x}(k+1) - \boldsymbol{x}(k) \rangle + L\|\boldsymbol{x}(k+1) - \boldsymbol{x}(k)\|^2\right)$$

$$\leq r_k + \frac{1}{L}\langle \nabla_{\boldsymbol{x}} C(\boldsymbol{x}(k); \boldsymbol{\theta} - \boldsymbol{\delta}(W-k)), \boldsymbol{x}^* - \boldsymbol{x}(k) \rangle$$

$$\quad - \frac{1}{L}\left(C(\boldsymbol{x}(k+1); \boldsymbol{\theta}) - C(\boldsymbol{x}(k); \boldsymbol{\theta}) - \frac{h^2}{2L}\|\boldsymbol{\delta}(W-k)\|^2\right)$$

$$\leq r_k + \frac{1}{L}\langle \nabla_{\boldsymbol{x}} C(\boldsymbol{x}(k); \boldsymbol{\theta} - \boldsymbol{\delta}(W-k)), \boldsymbol{x}^* - \boldsymbol{x}(k) \rangle$$

$$\quad - \frac{1}{L}\left(C(\boldsymbol{x}^*; \boldsymbol{\theta}) - C(\boldsymbol{x}(k); \boldsymbol{\theta})\right) + \frac{h^2}{2L^2}\|\boldsymbol{\delta}(W-k)\|^2$$

$$= r_k - \frac{1}{L}\left(C(\boldsymbol{x}^*; \boldsymbol{\theta}) - C(\boldsymbol{x}(k); \boldsymbol{\theta}) + \langle \nabla_{\boldsymbol{x}} C(\boldsymbol{x}(k); \boldsymbol{\theta}-), \boldsymbol{x}(k) - \boldsymbol{x}^* \rangle\right)$$

$$\quad + \frac{h^2}{2L^2}\|\boldsymbol{\delta}(W-k)\|^2$$

$$\leq r_k - \frac{1}{L}\left(\frac{\alpha}{4}\|\boldsymbol{x}(k) - \boldsymbol{x}^*\|^2 - \frac{h^2}{\alpha}\|\boldsymbol{\delta}(W-k)\|^2\right) + \frac{h^2}{2L^2}\|\boldsymbol{\delta}(W-k)\|^2$$

$$= \rho r_k + \frac{\zeta}{L}\|\boldsymbol{\delta}(W-k)\|^2$$

which completes the proof of (13).

*Step 3: completing the proof by* (13) *and* (12).
By summing (13) over $k = 0, \ldots, W - 2$, we obtain

$$r_{W-1} \leq \rho^{W-1} r_0 + \frac{\zeta}{L} \left( \|\boldsymbol{\delta}(2)\|^2 + \rho \|\boldsymbol{\delta}(3)\|^2 + \cdots + \rho^{W-2} \|\boldsymbol{\delta}(W)\|^2 \right)$$

$$\leq \rho^{W-1} \frac{2}{\alpha} (C(\boldsymbol{x}(0); \boldsymbol{\theta}) - C(\boldsymbol{x}^*; \boldsymbol{\theta})) + \frac{\zeta}{L} \sum_{k=2}^{W} \rho^{k-2} \|\boldsymbol{\delta}(k)\|^2$$

By (12), we obtain the regret bound in Theorem 1:

$$\text{Reg}(RHIG) \leq L\rho \left( \rho^{W-1} \frac{2}{\alpha} (C(\boldsymbol{x}(0); \boldsymbol{\theta}) - C(\boldsymbol{x}^*; \boldsymbol{\theta})) + \frac{\zeta}{L} \sum_{k=2}^{W} \rho^{k-2} \|\boldsymbol{\delta}(k)\|^2 \right) + \zeta \|\boldsymbol{\delta}(1)\|^2$$

$$= \frac{2L}{\alpha} \rho^W \text{Reg}(\phi) + \zeta \sum_{k=1}^{W} \rho^{k-1} \|\boldsymbol{\delta}(k)\|^2$$

$$= \frac{2L}{\alpha} \rho^W \text{Reg}(\phi) + \zeta \sum_{k=1}^{\min(W,T)} \rho^{k-1} \|\boldsymbol{\delta}(k)\|^2 + \zeta \mathbb{1}_{(W>T)} \sum_{k=T+1}^{W} \rho^{k-1} \|\boldsymbol{\delta}(T)\|^2$$

$$= \frac{2L}{\alpha} \rho^W \text{Reg}(\phi) + \zeta \sum_{k=1}^{\min(W,T)} \rho^{k-1} \|\boldsymbol{\delta}(k)\|^2 + \zeta \mathbb{1}_{(W>T)} \frac{\rho^T - \rho^W}{1 - \rho} \|\boldsymbol{\delta}(T)\|^2$$

where we used the fact that $\|\boldsymbol{\delta}(k)\| = \|\boldsymbol{\delta}(T)\|$ when $k > T$.

## D   Proofs of Theorem 2, Corollary 1 and the claimed properties of the regret bound in Section 4

In this section, we provide a dynamic regret bound for the restarted OGD initialization rule in Section 4, based on which we prove Corollary 1. To achieve this, we will first establish a static regret bound for OGD initialization (6). The proof is inspired by [34].

For notational simplicity, we slightly abuse the notation and let $x_t$ denote $x_t(0)$ generated by OGD. Further, by the definition of the prediction errors $\delta_{t-1}(W)$ for $W \geq 1$, we can write the initialization rule (6) as the following, which can be interpreted as OGD with inexact gradients:

$$x_t = \Pi_{\mathbb{X}}[x_{t-1} - \xi_t \nabla_x f(x_{t-1}; \theta_{t-1} - \delta_{t-1}(\min(W, T)))], \quad t \geq 2; \tag{15}$$

and $x_1 = x_0$. Here, we used the facts that $\theta_{t-1|t-W-1} = \theta_{t-1} - \delta_{t-1}(W)$ and $\delta_{t-1}(W) = \delta_{t-1}(T)$ for $W > T$.

### D.1   Static regret bound for OGD with inexact gradients

In this section, we consider the OGD with inexact gradients (15) with diminishing stepsize $\xi_t = \frac{4}{\alpha t}$ for $t \geq 1$. We will prove its static regret bound below.

**Theorem 6** (Static regret of OGD with inexact gradients)**.** *Consider the OGD with inexact gradients* (15) *with diminishing stepsize* $\xi_t = \frac{4}{\alpha t}$ *for* $t \geq 1$ *and any* $x_0$. *Then, for* $z^* = \arg\min_{z \in \mathbb{X}} \sum_{t=1}^{T} f(z; \theta_t)$, *we have the following static regret bound:*

$$\sum_{t=1}^{T} [f(x_t; \theta_t) - f(z^*; \theta_t)] \leq \frac{2G^2}{\alpha} \log(T+1) + \sum_{t=1}^{T} \frac{h^2}{\alpha} \|\delta_t(\min(W, T))\|^2$$

*Further, the total switching cost can be bounded by:*

$$\sum_{t=1}^{T} d(x_t, x_{t-1}) \leq \frac{16G^2 \beta}{\alpha^2}$$

*Proof.* Firstly, we prove the static regret bound. Define $q_t = \|x_t - z^*\|^2$. Then, for $t \geq 1$, we have the following.

$$
\begin{aligned}
q_{t+1} =& \|x_{t+1} - z^*\|^2 \leq \|x_t - \xi_{t+1}\nabla_x f(x_t; \theta_t - \delta_t(\min(W,T))) - z^*\|^2 \\
=& q_t + \xi_{t+1}^2 \|\nabla_x f(x_t; \theta_t - \delta_t(\min(W,T)))\|^2 - 2\xi_{t+1}\langle x_t - z^*, \nabla_x f(x_t; \theta_t - \delta_t(\min(W,T)))\rangle \\
\leq& q_t + \xi_{t+1}^2 G^2 - 2\xi_{t+1}\langle x_t - z^*, \nabla_x f(x_t; \theta_t)\rangle \\
& - 2\xi_{t+1}\langle x_t - z^*, \nabla_x f(x_t; \theta_t - \delta_t(\min(W,T))) - \nabla_x f(x_t; \theta_t)\rangle
\end{aligned}
$$

where the last inequality uses Assumption 3(i). By rearranging terms, we obtain

$$
\begin{aligned}
\langle x_t - z^*, \nabla_x f(x_t; \theta_t)\rangle \leq & \frac{q_t - q_{t+1}}{2\xi_{t+1}} + \frac{\xi_{t+1}}{2}G^2 \\
& - \langle x_t - z^*, \nabla_x f(x_t; \theta_t - \delta_t(\min(W,T))) - \nabla_x f(x_t; \theta_t)\rangle
\end{aligned}
\tag{16}
$$

By the strong convexity of $f(x; \theta_t)$, we have $f(z^*; \theta_t) \geq f(x_t; \theta_t) + \langle z^* - x_t, \nabla_x f(x_t; \theta_t)\rangle + \frac{\alpha}{2}\|z^* - x_t\|^2$. By rearranging terms and by (16), we obtain

$$
\begin{aligned}
& f(x_t; \theta_t) - f(z^*; \theta_t) \leq \langle x_t - z^*, \nabla_x f(x_t; \theta_t)\rangle - \frac{\alpha}{2}\|z^* - x_t\|^2 \\
\leq & \frac{q_t - q_{t+1}}{2\xi_{t+1}} + \frac{\xi_{t+1}}{2}G^2 - \langle x_t - z^*, \nabla_x f(x_t; \theta_t - \delta_t(\min(W,T))) - \nabla_x f(x_t; \theta_t)\rangle - \frac{\alpha}{2}\|z^* - x_t\|^2 \\
\leq & \frac{q_t - q_{t+1}}{2\xi_{t+1}} + \frac{\xi_{t+1}}{2}G^2 + \|x_t - z^*\|\|\nabla_x f(x_t; \theta_t - \delta_t(\min(W,T))) - \nabla_x f(x_t; \theta_t)\| - \frac{\alpha}{2}\|z^* - x_t\|^2 \\
\leq & \frac{q_t - q_{t+1}}{2\xi_{t+1}} + \frac{\xi_{t+1}}{2}G^2 + \frac{1}{\alpha}\|\nabla_x f(x_t; \theta_t - \delta_t(\min(W,T))) - \nabla_x f(x_t; \theta_t)\|^2 - \frac{\alpha}{4}\|z^* - x_t\|^2 \\
\leq & \frac{q_t - q_{t+1}}{2\xi_{t+1}} + \frac{\xi_{t+1}}{2}G^2 + \frac{h^2}{\alpha}\|\delta_t(\min(W,T))\|^2 - \frac{\alpha}{4}q_t
\end{aligned}
$$

where we used $ab \leq \frac{\epsilon}{2}a^2 + \frac{1}{2\epsilon}b^2$ for any $a, b \in \mathbb{R}$ and any $\epsilon > 0$ in the second last inequality and Assumption 2 in the last inequality. By summing over $t = 1, \ldots, T$, we obtain

$$
\begin{aligned}
\sum_{t=1}^{T}[f(x_t; \theta_t) - f(z^*; \theta_t)] \leq & \sum_{t=2}^{T}\left(\frac{1}{2\xi_{t+1}} - \frac{1}{2\xi_t} - \frac{\alpha}{4}\right)q_t + \left(\frac{1}{2\xi_2} - \frac{\alpha}{4}\right)q_1 - \frac{1}{\xi_{T+1}}q_{T+1} \\
& + \sum_{t=1}^{T}\frac{\xi_{t+1}}{2}G^2 + \sum_{t=1}^{T}\frac{h^2}{\alpha}\|\delta_t(\min(W,T))\|^2 \\
\leq & \log(T+1)\frac{2G^2}{\alpha} + \sum_{t=1}^{T}\frac{h^2}{\alpha}\|\delta_t(\min(W,T))\|^2
\end{aligned}
$$

which completes the proof of the static regret bound.

Next, we bound the switching costs. By Assumption 3(ii), we have

$$
\begin{aligned}
\sum_{t=1}^{T}d(x_t, x_{t-1}) \leq & \sum_{t=1}^{T}\frac{\beta}{2}\|x_t - x_{t-1}\|^2 \\
\leq & \sum_{t=1}^{T}\frac{\beta}{2}\|\xi_t \nabla_x f(x_{t-1}; \theta_{t-1} - \delta_{t-1}(\min(W,T)))\|^2 \\
\leq & \frac{\beta G^2}{2}\sum_{t=1}^{T}\xi_t^2 \leq \frac{16\beta G^2}{\alpha^2}
\end{aligned}
$$

$\square$

## D.2 Proof of Theorem 2: dynamic regret bound for restarted OGD with inexact gradients

We denote the set of stages in epoch $k$ as $\mathcal{T}_k = \{k\Delta + 1, \ldots, \min(k\Delta + \Delta, T)\}$ for $k = 0, \ldots, \lceil T/\Delta \rceil - 1$. We introduce $z_k^* = \arg\min_{z \in \mathbb{X}} \sum_{t \in \mathcal{T}_k}[f(z; \theta_t)]$ for all $k$; $y_t^* = \arg\min_{x_t \in \mathbb{X}} f(x_t; \theta_t)$ for all $t$; and $\boldsymbol{x}^* = \arg\min_{\boldsymbol{x} \in \mathbb{X}^T} \sum_{t=1}^T [f(x_t; \theta_t) + d(x_t, x_{t-1})]$. The dynamic regret of the restarted OGD with inexact gradients can be bounded as follows.

$$
\begin{aligned}
\text{Reg}(OGD) &= \sum_{t=1}^T [f(x_t; \theta_t) + d(x_t, x_{t-1})] - \sum_{t=1}^T [f(x_t^*; \theta_t) + d(x_t^*, x_{t-1}^*)] \\
&\leq \sum_{t=1}^T [f(x_t; \theta_t) + d(x_t, x_{t-1})] - \sum_{t=1}^T [f(x_t^*; \theta_t)] \\
&\leq \sum_{t=1}^T [f(x_t; \theta_t) + d(x_t, x_{t-1})] - \sum_{t=1}^T [f(y_t^*; \theta_t)] \\
&= \sum_{k=0}^{\lceil T/\Delta \rceil - 1} \sum_{t \in \mathcal{T}_k} [f(x_t; \theta_t) + d(x_t, x_{t-1}) - f(y_t^*; \theta_t)] \\
&= \sum_{k=0}^{\lceil T/\Delta \rceil - 1} \sum_{t \in \mathcal{T}_k} [f(x_t; \theta_t) - f(z_k^*; \theta_t)] + \sum_{k=0}^{\lceil T/\Delta \rceil - 1} \sum_{t \in \mathcal{T}_k} d(x_t, x_{t-1}) \\
&\quad + \sum_{k=0}^{\lceil T/\Delta \rceil - 1} \sum_{t \in \mathcal{T}_k} [f(z_k^*; \theta_t) - f(y_t^*; \theta_t)] \\
&\leq \lceil T/\Delta \rceil \log(\Delta + 1) \frac{2G^2}{\alpha} + \frac{h^2}{\alpha} \|\boldsymbol{\delta}(\min(W, T))\|^2 + \lceil T/\Delta \rceil \frac{16\beta G^2}{\alpha^2} \\
&\quad + \sum_{k=0}^{\lceil T/\Delta \rceil - 1} \sum_{t \in \mathcal{T}_k} [f(z_k^*; \theta_t) - f(y_t^*; \theta_t)]
\end{aligned}
$$

where the first inequality uses Assumption 3, the second inequality uses the optimality of $y_t^*$, the last inequality uses Theorem 6 and the fact that the OGD considered here restarts at the beginning of each epoch $k$ and repeats the stepsizes defined in Theorem 6, thus satisfying the static regret bound and the switching cost bound in Theorem 6 within each epoch.

Now, it suffices to bound $\sum_{k=0}^{\lceil T/\Delta \rceil - 1} \sum_{t \in \mathcal{T}_k} [f(z_k^*; \theta_t) - f(y_t^*; \theta_t)]$. By the optimality of $z_k^*$, we have:

$$
\sum_{k=0}^{\lceil T/\Delta \rceil - 1} \sum_{t \in \mathcal{T}_k} [f(z_k^*; \theta_t) - f(y_t^*; \theta_t)] \leq \sum_{k=0}^{\lceil T/\Delta \rceil - 1} \sum_{t \in \mathcal{T}_k} [f(y_{k\Delta+1}^*; \theta_t) - f(y_t^*; \theta_t)]. \tag{17}
$$

We define $V^k = \sum_{t \in \mathcal{T}_k} \sup_{x \in \mathbb{X}} |f(x; \theta_t) - f(x; \theta_{t-1})|$. Then, for any $t \in \mathcal{T}_k$, we obtain

$$
\begin{aligned}
f(y_{k\Delta+1}^*; \theta_t) - f(y_t^*; \theta_t) &= f(y_{k\Delta+1}^*; \theta_t) - f(y_{k\Delta+1}^*; \theta_{k\Delta+1}) + f(y_{k\Delta+1}^*; \theta_{k\Delta+1}) - f(y_t^*; \theta_{k\Delta+1}) \\
&\quad + f(y_t^*; \theta_{k\Delta+1}) - f(y_t^*; \theta_t) \\
&\leq V^k + 0 + V^k = 2V^k
\end{aligned}
$$

By summing over $t \in \mathcal{T}_k$ and $k = 0, \ldots, \lceil T/\Delta \rceil - 1$ and by the inequality (17), we obtain

$$
\sum_{k=0}^{\lceil T/\Delta \rceil - 1} \sum_{t \in \mathcal{T}_k} [f(z_k^*; \theta_t) - f(y_t^*; \theta_t)] \leq \sum_{k=0}^{\lceil T/\Delta \rceil - 1} \sum_{t \in \mathcal{T}_k} 2V^k = \sum_{k=0}^{\lceil T/\Delta \rceil - 1} 2\Delta V^k = 2\Delta V_T
$$

Combining the bounds above yields the desired bound on the dynamic regret of OGD below by letting $\Delta = \lceil \sqrt{2T/V_T} \rceil$:

$$
\text{Reg}(OGD) \leq \lceil T/\Delta \rceil \log(\Delta + 1) \frac{2G^2}{\alpha} + \frac{h^2}{\alpha} \|\boldsymbol{\delta}(\min(W, T))\|^2 + \lceil T/\Delta \rceil \frac{16\beta G^2}{\alpha^2} + 2\Delta V_T
$$

$$\leq \left( \sqrt{\frac{V_T T}{2}} + 1 \right) \log(2 + \sqrt{2T/V_T}) \frac{2G^2}{\alpha} + \frac{h^2}{\alpha} \|\boldsymbol{\delta}(\min(W,T))\|^2$$

$$+ \left( \sqrt{\frac{V_T T}{2}} + 1 \right) \frac{16\beta G^2}{\alpha^2} + 2(\sqrt{2V_T T} + V_T)$$

$$\leq (\sqrt{V_T T/2} + 1) \log(2 + \sqrt{2T/V_T}) \left( \frac{2G^2}{\alpha} + \frac{16\beta G^2}{\alpha^2} + 2(2 + \sqrt{2}) \right) + \frac{h^2}{\alpha} \|\boldsymbol{\delta}(\min(W,T))\|^2$$

$$\leq \sqrt{2V_T T} \log(2 + \sqrt{2T/V_T}) \left( \frac{2G^2}{\alpha} + \frac{16\beta G^2}{\alpha^2} + 2(2 + \sqrt{2}) \right) + \frac{h^2}{\alpha} \|\boldsymbol{\delta}(\min(W,T))\|^2$$

$$\leq \sqrt{V_T T} \log(1 + \sqrt{T/V_T}) \left( \frac{4\sqrt{2}G^2}{\alpha} + \frac{32\sqrt{2}\beta G^2}{\alpha^2} + 8(1 + \sqrt{2}) \right) + \frac{h^2}{\alpha} \|\boldsymbol{\delta}(\min(W,T))\|^2$$

$$\leq \sqrt{V_T T} \log(1 + \sqrt{T/V_T}) \left( \frac{4\sqrt{2}G^2}{\alpha} + \frac{32\sqrt{2}\beta G^2}{\alpha^2} + 20 \right) + \frac{h^2}{\alpha} \|\boldsymbol{\delta}(\min(W,T))\|^2$$

where we used the facts that $\lceil x \rceil \leq x + 1$, $1 \leq V_T \leq T$, $T > 2$, $\log(2 + \sqrt{2T/V_T}) \leq 2\log(1 + \sqrt{T/V_T})$, and $8(1 + \sqrt{2}) < 20$.

### D.3 Proof of Corollary 1

The proof is straightforward by substituting restarted OGD's regret bound in Theorem 2 into the general regret bound in Theorem 1, that is,

$$\text{Reg}(RHIG) \leq \rho^W \frac{2L}{\alpha} C_1 \sqrt{V_T T} \log(1 + \sqrt{T/V_T})$$

$$+ \frac{2L}{\alpha} \frac{h^2}{\alpha} \rho^W \|\boldsymbol{\delta}(\min(W,T))\|^2 + \sum_{k=1}^{\min(W,T)} \zeta \rho^{k-1} \|\boldsymbol{\delta}(k)\|^2 + \mathbb{1}_{(W>T)} \frac{\rho^T - \rho^W}{1 - \rho} \zeta \|\boldsymbol{\delta}(T)\|^2.$$

### D.4 Proofs of the monotonicity claims in the discussion of Corollary 1.

In Section 4, when discussing **Choices of** $W$, we claim that "Part I increases with $V_T$ and Part II increases with the prediction errors. Further, as $W$ increases, Part I decreases but Part II increases." For completeness, we prove this claim below.

**Properties of Part I** $\rho^W \frac{2L}{\alpha} C_1 \sqrt{V_T T} \log(1 + \sqrt{T/V_T})$: Since $0 < \rho < 1$, it is straightforward that Part I monotonically decreases with $W$. Next, consider function $p(x) = x \log(1 + \frac{b}{x})$ for $x, b > 0$. Since $p'(x) = \frac{x}{x+b} - 1 - \log(\frac{x}{x+b}) \geq 0$ by $y - 1 \geq \log(y)$ for any $y > 0$, function $p(x)$ monotonically increases with $x$. Therefore, for any fixed $W$, Part I monotonically increases with $\sqrt{V_T}$ and thus $V_T$.

**Properties of Part II** $\frac{2Lh^2}{\alpha^2} \rho^W \|\boldsymbol{\delta}(\min(W,T))\|^2 + \sum_{k=1}^{\min(W,T)} \zeta \rho^{k-1} \|\boldsymbol{\delta}(k)\|^2 + \mathbb{1}_{(W>T)} \frac{\rho^T - \rho^W}{1-\rho} \zeta \|\boldsymbol{\delta}(T)\|^2$: It is straightforward that Part II monotonically increases with $\{\|\boldsymbol{\delta}(k)\|^2\}_{k=1}^W$. Next, we discuss the monotonicty with respect to $W$. We first consider $W \leq T$. In this case, Part II is equal to Part II$(W) := \frac{2Lh^2}{\alpha^2} \rho^W \|\boldsymbol{\delta}(W)\|^2 + \sum_{k=1}^W \zeta \rho^{k-1} \|\boldsymbol{\delta}(k)\|^2$. Notice that

$$\text{Part II}(W) - \text{Part II}(W-1) = \frac{2Lh^2}{\alpha^2} \rho^W \|\boldsymbol{\delta}(W)\|^2 + \zeta \rho^{W-1} \|\boldsymbol{\delta}(W)\|^2 - \frac{2Lh^2}{\alpha^2} \rho^{W-1} \|\boldsymbol{\delta}(W-1)\|^2$$

$$\geq \left( \frac{2Lh^2}{\alpha^2} \rho + \zeta - \frac{2Lh^2}{\alpha^2} \right) \rho^{W-1} \|\boldsymbol{\delta}(W-1)\|^2$$

$$= \left( \frac{h^2}{2\alpha} + \frac{h^2}{2L} \right) \rho^{W-1} \|\boldsymbol{\delta}(W-1)\|^2 > 0$$

where we used $\|\boldsymbol{\delta}(W)\|^2 \geq \|\boldsymbol{\delta}(W-1)\|^2$, $\rho = 1 - \frac{\alpha}{4L}$, $\zeta = \frac{h^2}{\alpha} + \frac{h^2}{2L}$. Therefore, Part II is monotonically increasing with $W$ for $W \leq T$. Besides, we consider $W > T$. In this case, Part II is

equal to Part II$(W) := \frac{2Lh^2}{\alpha^2}\rho^W\|\boldsymbol{\delta}(T))\|^2 + \sum_{k=1}^{T}\zeta\rho^{k-1}\|\boldsymbol{\delta}(k)\|^2 + \frac{\rho^T - \rho^W}{1-\rho}\zeta\|\boldsymbol{\delta}(T)\|^2$. Notice that, when $W > T$, we have

$$
\begin{aligned}
\text{Part II}(W) - \text{Part II}(W-1) &= \frac{2Lh^2}{\alpha^2}(\rho^W - \rho^{W-1})\|\boldsymbol{\delta}(T)\|^2 + \frac{\rho^{W-1} - \rho^W}{1-\rho}\zeta\|\boldsymbol{\delta}(T)\|^2 \\
&= \left(\frac{2Lh^2}{\alpha^2}(\rho - 1) + \zeta\right)\rho^{W-1}\|\boldsymbol{\delta}(T)\|^2 \\
&= \left(\frac{h^2}{2\alpha} + \frac{h^2}{2L}\right)\rho^{W-1}\|\boldsymbol{\delta}(W-1)\|^2 > 0
\end{aligned}
$$

In conclusion, Part II increases with $W$ for $W \geq 1$.

# E  Analysis on the special case in Section 4

## E.1  Proof of Corollary 2

For notational simplicity, let $R(W)$ denote the regret bound in Corollary 1 given lookahead horizon $W$, i.e.

$$
\begin{aligned}
R(W) = &\rho^W \frac{2L}{\alpha}C_1\sqrt{V_T T}\log(1 + \sqrt{T/V_T}) \\
&+ \frac{2L}{\alpha}\frac{h^2}{\alpha}\rho^W\|\boldsymbol{\delta}(\min(W,T))\|^2 + \sum_{k=1}^{\min(W,T)}\zeta\rho^{k-1}\|\boldsymbol{\delta}(k)\|^2 + \mathbb{1}_{(W>T)}\frac{\rho^T - \rho^W}{1-\rho}\zeta\|\boldsymbol{\delta}(T)\|^2.
\end{aligned}
$$

We will show that $R(W) \leq R(W-1)$ for $W \geq 1$. Firstly, we consider $W \leq T$. In this case, we have $R(W) = \rho^W\frac{2L}{\alpha}C_1\sqrt{V_T T}\log(1 + \sqrt{T/V_T}) + \frac{2L}{\alpha}\frac{h^2}{\alpha}\rho^W\|\boldsymbol{\delta}(W)\|^2 + \sum_{k=1}^{W}\zeta\rho^{k-1}\|\boldsymbol{\delta}(k)\|^2$. Notice that

$$
\begin{aligned}
R(W) - R(W-1) =& (\rho^W - \rho^{W-1})\frac{2L}{\alpha}C_1\sqrt{V_T T}\log(1 + \sqrt{T/V_T}) + \frac{2L}{\alpha}\frac{h^2}{\alpha}\rho^W\|\boldsymbol{\delta}(W)\|^2 \\
&+ \zeta\rho^{W-1}\|\boldsymbol{\delta}(W)\|^2 - \frac{2L}{\alpha}\frac{h^2}{\alpha}\rho^{W-1}\|\boldsymbol{\delta}(W-1)\|^2 \\
\leq& \rho^{W-1}\left[\left(\frac{2L}{\alpha}\frac{h^2}{\alpha}\rho + \zeta\right)\|\boldsymbol{\delta}(W)\|^2 - (1-\rho)\frac{2L}{\alpha}C_1\sqrt{V_T T}\log(1 + \sqrt{T/V_T})\right] \\
\leq& 0
\end{aligned}
$$

when the following condition holds for any $W \leq T$.

$$
\left(\frac{2L}{\alpha}\frac{h^2}{\alpha}\rho + \zeta\right)\|\boldsymbol{\delta}(W)\|^2 \leq (1-\rho)\frac{2L}{\alpha}C_1\sqrt{V_T T}\log(1 + \sqrt{T/V_T}) \tag{18}
$$

Next, we consider $W > T$. In this case, we have $R(W) = \rho^W\frac{2L}{\alpha}C_1\sqrt{V_T T}\log(1 + \sqrt{T/V_T}) + \frac{2L}{\alpha}\frac{h^2}{\alpha}\rho^W\|\boldsymbol{\delta}(T)\|^2 + \sum_{k=1}^{T}\zeta\rho^{k-1}\|\boldsymbol{\delta}(k)\|^2 + \frac{\rho^T - \rho^W}{1-\rho}\zeta\|\boldsymbol{\delta}(T)\|^2$. Therefore,

$$
\begin{aligned}
R(W) - R(W-1) =& (\rho^W - \rho^{W-1})\frac{2L}{\alpha}C_1\sqrt{V_T T}\log(1 + \sqrt{T/V_T}) + \frac{2L}{\alpha}\frac{h^2}{\alpha}(\rho^W - \rho^{W-1})\|\boldsymbol{\delta}(T)\|^2 \\
&+ \frac{\rho^{W-1} - \rho^W}{1-\rho}\zeta\|\boldsymbol{\delta}(T)\|^2 \\
\leq& \rho^{W-1}\left(\zeta\|\boldsymbol{\delta}(T)\|^2 - (1-\rho)\frac{2L}{\alpha}C_1\sqrt{V_T T}\log(1 + \sqrt{T/V_T})\right) \\
\leq& 0
\end{aligned}
$$

given the condition (18).

In conclusion, we have $R(W) \leq R(W-1)$ for $W \geq 1$ and the $R(W)$ is minimized by letting $W \to +\infty$. Further, when $W \to +\infty$, we have the following bound.

$$
\lim_{W \to +\infty} R(W) = \sum_{k=1}^{T}\zeta\rho^{k-1}\|\boldsymbol{\delta}(k)\|^2 + \frac{\rho^T}{1-\rho}\zeta\|\boldsymbol{\delta}(T)\|^2 \leq \frac{\zeta}{1-\rho}\sum_{k=1}^{T}\rho^{k-1}\|\boldsymbol{\delta}(k)\|^2
$$

## E.2 Proof of Theorem 3

Without loss of generality, we consider $n = 1$. It is straightforward to generalize the proof to $n > 1$ cases. The proof is based on constructing a special cost function where the lower bound holds.

Consider cost function $f(x_t; \theta_t) = \frac{\alpha}{2}(x_t^2 - 2\theta_t x_t)$ and $d(x_t, x_{t-1}) = \frac{\beta}{2}\|x_t - x_{t-1}\|^2$ on $\mathbb{X} = [-1/2, 1/2]$, where $l_f = \alpha$, $l_d = 2\beta$, $L = \alpha + 4\beta$ and $h = \alpha$. Let $\alpha > 1$ and $\frac{\beta}{\alpha} < 4 + 3\sqrt{2}$ so that $\rho_0 < 1/2$. Let $\theta_t \in \mathbb{X}$ for all $t$, then we have $G = \sup_{x \in \mathbb{X}} \|\alpha(x - \theta_t)\| = \alpha$. Let $x_0 = 0$.

Consider a random $\theta_t$:

$$\theta_t = \mu_t + e_1^t + \cdots + e_t^t, \quad \forall\, 1 \le t \le T,$$

where $e_\tau^t$ are independent variables across $1 \le t \le T$ and $1 \le \tau \le t$. Let the support of $e_\tau^t$ be $[-\frac{1}{8t}, \frac{1}{8t}]$ and let $\mu_t = (-1)^t \frac{1}{4}$, so $\theta_t \in \mathbb{X}$ is $\mathbb{X}$ and $\frac{1}{8} \le \theta_t \le \frac{3}{8}$ if $t$ is even and $\frac{-3}{8} \le \theta_t \le \frac{-1}{8}$ if $t$ is odd. Consider predictions at time $\tau$ as $\theta_{t|\tau} = \mu_t + e_1^t + \cdots + e_\tau^t$ for any $0 \le \tau \le t$. Therefore, $\delta_t(t - \tau) = e_{\tau+1}^t + \cdots + e_t^t$ and $\|\delta_t(t - \tau)\| \le \frac{t-\tau}{8t} \le 1/8$. According to our construction, we have that $V_T = \sum_{t=1}^{T} \sup_{x \in \mathbb{X}} \alpha\|(\theta_t - \theta_{t-1})x\| \ge \alpha T/8$; and $\|\boldsymbol{\delta}(k)\|^2 \le \frac{T}{64}$ for any $k \ge 1$. Then, it is straightforward to verify that the constructed cost functions and predictions satisfy $\sqrt{V_T T} \log(1 + \sqrt{T/V_T}) \ge \frac{2Lh^2\rho + \alpha^2\zeta}{2LC_1(1-\rho)\alpha}\|\boldsymbol{\delta}(k)\|^2$ for any $k \ge 1$.

Notice that knowing $\theta_{t|0}, \ldots, \theta_{t|\tau}$ is equivalent with knowing $\mu_t, e_1^t, \ldots, e_\tau^t$. Therefore, let filtration $\mathcal{F}_t$ denote all the information at $t$ provide by the predictions and the history, then $\mathcal{F}_t$ is generated by $\theta_1, \ldots, \theta_{t-1}$ and $\mu_s, e_1^s, \ldots, e_{t-1}^s$ for $s \ge t$. Notice that $\mathbb{E}[\theta_\tau \mid \mathcal{F}_t] = \theta_{\tau|t-1}$ for $\tau \ge t$. Besides, for any online algorithm $\mathcal{A}$, we have that $x_t^\mathcal{A}$ is measurable in $\mathcal{F}_t$.

Since $\theta_t \in \mathbb{X}$ for all $t$, it can be shown that the optimal solution $\boldsymbol{x}^* = \arg\min_{\boldsymbol{x} \in \mathbb{X}^T} C(\boldsymbol{x}; \boldsymbol{\theta})$ is an interior point of $\mathbb{X}^T$ and thus satisfies the first-order optimality condition $\boldsymbol{x}^* = A\boldsymbol{\theta}$, where $A$ is the inverse of the Hessian matrix of $C(\boldsymbol{x}; \boldsymbol{\theta})$. Equivalently, we have $x_t^* = \sum_{\tau=1}^{T} a_{t,\tau} \theta_\tau$. Further, Lemma 5 in [17] shows that $a_{t,\tau}^2 \ge c_2 \rho_0^{\tau-t}$ for $\tau \ge t$, where $c_2 = (\frac{\alpha}{\alpha+\beta})^2(1 - \sqrt{\rho_0})^2$.

Since $x_t^\mathcal{A}$ is measurable in $\mathcal{F}_t$, by the projection theory, we have

$$\mathbb{E}[\|x_t^\mathcal{A} - x_t^*\|^2] \ge \mathbb{E}[\|\mathbb{E}[x_t^* \mid \mathcal{F}_t] - x_t^*\|^2].$$

Notice that

$$\mathbb{E}[x_t^* \mid \mathcal{F}_t] = a_{t,1}\theta_1 + \cdots + a_{t,t-1}\theta_{t-1} + a_{t,t}\mathbb{E}[\theta_t \mid \mathcal{F}_t] + \cdots + a_{t,T}\mathbb{E}[\theta_T \mid \mathcal{F}_t]$$
$$= a_{t,1}\theta_1 + \cdots + a_{t,t-1}\theta_{t-1} + a_{t,t}\theta_{t|t-1} + a_{t,T}\theta_{T|t-1}$$

Therefore,

$$\mathbb{E}[\|\mathbb{E}[x_t^* \mid \mathcal{F}_t] - x_t^*\|^2] = \mathbb{E}[\|a_{t,t}\delta_t(1) + \cdots + a_{t,T}\delta_T(T - t + 1)\|^2]$$
$$= a_{t,t}^2\mathbb{E}[\|\delta_t(1)\|^2] + \cdots + a_{t,T}^2\mathbb{E}[\|\delta_T(T - t + 1)\|^2]$$
$$\ge c_2(\mathbb{E}[\|\delta_t(1)\|^2] + \cdots + \rho_0^{T-t}\mathbb{E}[\|\delta_T(T - t + 1)\|^2])$$

where we used the independence among the prediction errors and $a_{t,\tau}^2 \ge c_2 \rho_0^{\tau-t}$ for $\tau \ge t$.

Summing over $t$ leads to the following.

$$\sum_{t=1}^{T}\mathbb{E}[\|x_t^\mathcal{A} - x_t^*\|^2] \ge \sum_{t=1}^{T}\mathbb{E}[\|\mathbb{E}[x_t^* \mid \mathcal{F}_t] - x_t^*\|^2]$$
$$\ge \sum_{t=1}^{T}(a_{t,t}^2\mathbb{E}[\|\delta_t(1)\|^2] + \cdots + a_{t,T}^2\mathbb{E}[\|\delta_T(T - t + 1)\|^2])$$
$$\ge \sum_{t=1}^{T}c_2\sum_{k=1}^{T-t+1}\rho_0^{k-1}\mathbb{E}[\|\delta_{k+t-1}(k)\|^2]$$
$$= c_2\sum_{k=1}^{T}\rho_0^{k-1}\sum_{t=1}^{T+1-k}\mathbb{E}[\|\delta_{k+t-1}(k)\|^2]$$

$$= c_2 \sum_{k=1}^{T} \rho_0^{k-1} \sum_{t=1}^{T} \mathbb{E}[\|\delta_t(k)\|^2] - c_2 \sum_{k=1}^{T} \rho_0^{k-1} \sum_{t=1}^{k-1} \mathbb{E}[\|\delta_t(k)\|^2]$$

$$= c_2 \sum_{k=1}^{T} \rho_0^{k-1} \sum_{t=1}^{T} \mathbb{E}[\|\delta_t(k)\|^2] - c_2 \sum_{k=1}^{T} \rho_0^{k-1} \sum_{t=1}^{k-1} \mathbb{E}[\|\delta_t(t)\|^2]$$

$$= c_2 \sum_{k=1}^{T} \rho_0^{k-1} \sum_{t=1}^{T} \mathbb{E}[\|\delta_t(k)\|^2] - c_2 \sum_{t=1}^{T-1} \mathbb{E}[\|\delta_t(t)\|^2] \sum_{k=t+1}^{T} \rho_0^{k-1}$$

$$\geq c_2 \sum_{k=1}^{T} \rho_0^{k-1} \sum_{t=1}^{T} \mathbb{E}[\|\delta_t(k)\|^2] - c_2 \sum_{k=1}^{T-1} \mathbb{E}[\|\delta_k(k)\|^2] \frac{\rho_0^k}{1-\rho_0}$$

$$\geq c_2 \sum_{k=1}^{T} \rho_0^{k-1} \sum_{t=1}^{T} \mathbb{E}[\|\delta_t(k)\|^2] \frac{1-2\rho_0}{1-\rho_0}$$

where we used $\delta_t(k) = \delta_t(t)$ for $k \geq t$ in the third equality and change the counting index from $t$ to $k$ in the second last inequality.

By strong convexity, we have $\mathbb{E}[\text{Reg}(\mathcal{A})] \geq \frac{\alpha}{2} \mathbb{E} \|x^{\mathcal{A}} - x^*\|^2 \geq c_2 \frac{\alpha}{2} \frac{1-2\rho_0}{1-\rho_0} \sum_{k=1}^{T} \rho_0^{k-1} \mathbb{E}[\|\boldsymbol{\delta}(k)\|^2]$. Therefore, there must exist a scenario such that $\text{Reg}(\mathcal{A}) \geq c_2 \frac{\alpha}{2} \frac{1-2\rho_0}{1-\rho_0} \sum_{k=1}^{T} \rho_0^{k-1} \|\boldsymbol{\delta}(k)\|^2$. Since $h = \alpha$ in our construction, we complete the proof.

# F  Stochastic Regret Analysis

## F.1  Proof of Theorem 4

By taking expectation on both sides of the regret bound in Theorem 1, we have

$$\mathbb{E}[\text{Reg}(RHIG)] \leq \frac{2L}{\alpha} \rho^W \mathbb{E}[\text{Reg}(\phi)] + \zeta \sum_{k=1}^{\min(W,T)} \rho^{k-1} \mathbb{E}[\|\boldsymbol{\delta}(k)\|^2] + \mathbb{1}_{(W>T)} \frac{\rho^T - \rho^W}{1-\rho} \zeta \mathbb{E}[\|\boldsymbol{\delta}(T)\|^2].$$

$$(19)$$

Therefore, it suffices to bound $\mathbb{E}[\|\boldsymbol{\delta}(k)\|^2]$ for $1 \leq k \leq T$. By $\boldsymbol{\delta}(k) = (\delta_1(k)^\top, \ldots, \delta_T(k)^\top)^\top$, $\delta_t(k) = \theta_t - \theta_{t|t-k} = P(0)e_t + \cdots + P(k-1)e_{t-k+1}$ for $k \leq t$ and $\delta_t(k) = \delta_t(t)$ for $k > t$, we have

$$\boldsymbol{\delta}(k) = M_k e, \quad 1 \leq k \leq T \tag{20}$$

where we define $e = (e_1^\top, \ldots, e_T^\top)^\top \in \mathbb{R}^{qT}$ and

$$M_k = \begin{bmatrix} P(0) & 0 & \cdots & \cdots & \cdots & 0 \\ P(1) & P(0) & \cdots & \cdots & \cdots & 0 \\ \vdots & \ddots & \ddots & \ddots & \ddots & 0 \\ P(k-1) & \cdots & P(1) & P(0) & \cdots & 0 \\ \vdots & \ddots & & \ddots & \ddots & 0 \\ 0 & \cdots & P(k-1) & \cdots & P(1) & P(0) \end{bmatrix}.$$

Let $R_e$ denote the covariance matrix of $e$, i.e.

$$R_e = \begin{bmatrix} R_e & 0 & \cdots & 0 \\ 0 & R_e & \cdots & 0 \\ \vdots & \ddots & \ddots & \vdots \\ 0 & \cdots & \cdots & R_e \end{bmatrix}.$$

Then, for $k \leq T$, we have

$$\mathbb{E}[\|\boldsymbol{\delta}(k)\|^2] = \mathbb{E}[e^\top M_k^\top M_k e] = \mathbb{E}[\text{tr}(ee^\top M_k^\top M_k)]$$

$$= \operatorname{tr}\left(\boldsymbol{R_e}\boldsymbol{M}_k^\top \boldsymbol{M}_k\right)$$

$$\leq \|R_e\|\|\boldsymbol{M}_k\|_F^2 = \|R_e\| \sum_{t=0}^{k-1}(T-t)\|P(t)\|_F^2$$

where the first inequality is by $\operatorname{tr}(AB) \leq \|A\|\operatorname{tr}(B)$ for any symmetrix matrices $A, B$, and $\|\operatorname{diag}(R_e, \ldots, R_e)\| = \|R_e\|$ and $\operatorname{tr}(A^\top A) = \|A\|_F^2$ for any matrix $A$. In addition, for $k \geq T$, we have $\mathbb{E}[\|\boldsymbol{\delta}(k)\|^2] \leq \|R_e\| \sum_{t=0}^{T-1}(T-t)\|P(t)\|_F^2$. In conclusion, for any $k \geq 1$, we have

$$\mathbb{E}[\|\boldsymbol{\delta}(k)\|^2] \leq \|R_e\| \sum_{t=0}^{\min(k,T)-1}(T-t)\|P(t)\|_F^2 \tag{21}$$

When $W \leq T$, substituting the bounds on $\mathbb{E}[\|\boldsymbol{\delta}(k)\|^2]$ into (19) yields the bound on the expected regret below.

$$\mathbb{E}[\operatorname{Reg}(RHIG)] \leq \frac{2L}{\alpha}\rho^W \mathbb{E}[\operatorname{Reg}(\phi)] + \zeta \sum_{k=1}^{W}\rho^{k-1}\|R_e\|\sum_{t=0}^{k-1}(T-t)\|P(t)\|_F^2$$

$$= \frac{2L}{\alpha}\rho^W \mathbb{E}[\operatorname{Reg}(\phi)] + \zeta \sum_{t=0}^{W-1}\|R_e\|(T-t)\|P(t)\|_F^2 \sum_{k=t+1}^{W}\rho^{k-1}$$

$$= \frac{2L}{\alpha}\rho^W \mathbb{E}[\operatorname{Reg}(\phi)] + \zeta \sum_{t=0}^{W-1}\|R_e\|(T-t)\|P(t)\|_F^2 \frac{\rho^t - \rho^W}{1-\rho}$$

When $W \geq T$, substituting the bounds on $\mathbb{E}[\|\boldsymbol{\delta}(k)\|^2]$ into (19) yields the bound on the expected regret below.

$$\mathbb{E}[\operatorname{Reg}(RHIG)] \leq \frac{2L}{\alpha}\rho^W \mathbb{E}[\operatorname{Reg}(\phi)] + \zeta \sum_{k=1}^{T}\rho^{k-1}\|R_e\|\sum_{t=0}^{k-1}(T-t)\|P(t)\|_F^2$$

$$+ \zeta\frac{\rho^T - \rho^W}{1-\rho}\|R_e\|\sum_{t=0}^{T-1}(T-t)\|P(t)\|_F^2$$

$$= \frac{2L}{\alpha}\rho^W \mathbb{E}[\operatorname{Reg}(\phi)] + \zeta \sum_{t=0}^{T-1}\|R_e\|(T-t)\|P(t)\|_F^2 \left(\sum_{k=t+1}^{T}\rho^{k-1} + \frac{\rho^T - \rho^W}{1-\rho}\right)$$

$$= \frac{2L}{\alpha}\rho^W \mathbb{E}[\operatorname{Reg}(\phi)] + \zeta \sum_{t=0}^{T-1}\|R_e\|(T-t)\|P(t)\|_F^2 \frac{\rho^t - \rho^W}{1-\rho}$$

In conclusion, we have the regret bound for general $W \geq 0$ below.

$$\mathbb{E}[\operatorname{Reg}(RHIG)] \leq \frac{2L}{\alpha}\rho^W \operatorname{Reg}(\phi) + \zeta \sum_{t=0}^{\min(W,T)-1}\|R_e\|(T-t)\|P(t)\|_F^2 \frac{\rho^t - \rho^W}{1-\rho}$$

### F.2 Proof of Corollary 3

The proof is very similar to the proof of Theorem 2. Before the proof, we note that we cannot directly apply the expected regret bound in the literature [34] due to the major differences in the problem formulation as discussed below. Firstly, the expected regret definition considered in this paper is different from that in [34] because the true cost function parameter $\theta_t$ in our case is also random and taken expectation on, while the true cost function in [34] is deterministic and the expectation is only taken on the random gradient noises. Besides, [34] considers unbiased gradient estimation while our gradient estimation $\nabla_{x_t} f(x_t; \theta_{t|\tau})$ can be biased. Further, [34] considers independent gradient noises at each stage $t$, while our gradient noises are correlated due to the correlation among prediction errors. Therefore, we have to revise the original proof in [34] for a new regret bound for our setting.

Similar to the proof of Theorem 2, we denote the set of stages in epoch $k$ as $\mathcal{T}_k = \{k\Delta + 1, \ldots, \min(k\Delta + \Delta, T)\}$ for $k = 0, \ldots, \lceil T/\Delta \rceil - 1$; and introduce $z_k^* =$

$\arg\min_{z \in \mathbb{X}} \sum_{t \in \mathcal{T}_k} [f(z; \theta_t)]$, $y_t^* = \arg\min_{x \in \mathbb{X}} f(x; \theta_t)$, $\boldsymbol{x}^* = \arg\min_{\boldsymbol{x} \in \mathbb{X}^T} \sum_{t=1}^{T} [f(x_t; \theta_t) + d(x_t, x_{t-1})]$. Notice that $z_k^*, y_t^*, x_t^*$ are all random variables depending on $\boldsymbol{\theta}$. The expected dynamic regret of OGD can be bounded as follows.

$$\mathbb{E}[\text{Reg}(OGD)] = \sum_{t=1}^{T} \mathbb{E}[f(x_t; \theta_t) + d(x_t, x_{t-1})] - \sum_{t=1}^{T} \mathbb{E}[f(x_t^*; \theta_t) + d(x_t^*, x_{t-1}^*)]$$

$$\leq \sum_{t=1}^{T} \mathbb{E}[f(x_t; \theta_t) + d(x_t, x_{t-1})] - \sum_{t=1}^{T} \mathbb{E}[f(x_t^*; \theta_t)]$$

$$\leq \sum_{t=1}^{T} \mathbb{E}[f(x_t; \theta_t) + d(x_t, x_{t-1})] - \sum_{t=1}^{T} \mathbb{E}[f(y_t^*; \theta_t)]$$

$$= \sum_{k=0}^{\lceil T/\Delta \rceil - 1} \sum_{t \in \mathcal{T}_k} \mathbb{E}[f(x_t; \theta_t) + d(x_t, x_{t-1}) - f(y_t^*; \theta_t)]$$

$$= \sum_{k=0}^{\lceil T/\Delta \rceil - 1} \sum_{t \in \mathcal{T}_k} \mathbb{E}[f(x_t; \theta_t) - f(z_k^*; \theta_t)] + \sum_{k=0}^{\lceil T/\Delta \rceil - 1} \sum_{t \in \mathcal{T}_k} \mathbb{E}[d(x_t, x_{t-1})]$$

$$+ \sum_{k=0}^{\lceil T/\Delta \rceil - 1} \sum_{t \in \mathcal{T}_k} \mathbb{E}[f(z_k^*; \theta_t) - f(y_t^*; \theta_t)]$$

$$\leq \lceil T/\Delta \rceil \log(\Delta + 1) \frac{2G^2}{\alpha} + \frac{h^2}{\alpha} \mathbb{E} \|\boldsymbol{\delta}(\min(W, T))\|^2 + \lceil T/\Delta \rceil \frac{16\beta G^2}{\alpha^2}$$

$$+ \sum_{k=0}^{\lceil T/\Delta \rceil - 1} \sum_{t \in \mathcal{T}_k} \mathbb{E}[f(z_k^*; \theta_t) - f(y_t^*; \theta_t)]$$

where the first inequality uses Assumption 3, the second inequality uses the optimality of $y_t^*$, the last inequality follows from taking expectation on the regret bounds in Theorem 6 and the fact that the OGD considered here restarts at the beginning of each epoch $k$ and repeats the stepsizes defined in Theorem 6, thus satisfying the static regret bound and switching cost bound in Theorem 6 within each epoch.

Now, it suffices to bound $\sum_{k=0}^{\lceil T/\Delta \rceil - 1} \sum_{t \in \mathcal{T}_k} \mathbb{E}[f(z_k^*; \theta_t) - f(y_t^*; \theta_t)]$. By the optimality of $z_k^*$, we have that

$$\sum_{k=0}^{\lceil T/\Delta \rceil - 1} \sum_{t \in \mathcal{T}_k} \mathbb{E}[f(z_k^*; \theta_t) - f(y_t^*; \theta_t)] \leq \sum_{k=0}^{\lceil T/\Delta \rceil - 1} \sum_{t \in \mathcal{T}_k} \mathbb{E}[f(y_{k\Delta+1}^*; \theta_t) - f(y_t^*; \theta_t)].$$

We define $\mathbb{E}[V^k] = \sum_{t \in \mathcal{T}_k} \mathbb{E}[\sup_{x \in \mathbb{X}} |f(x; \theta_t) - f(x; \theta_{t-1})|]$.[11] Then, for any $t \in \mathcal{T}_k$, we obtain

$$\mathbb{E}[f(y_{k\Delta+1}^*; \theta_t) - f(y_t^*; \theta_t)]$$
$$= \mathbb{E}[f(y_{k\Delta+1}^*; \theta_t) - f(y_{k\Delta+1}^*; \theta_{k\Delta+1})] + \mathbb{E}[f(y_{k\Delta+1}^*; \theta_{k\Delta+1}) - f(y_t^*; \theta_{k\Delta+1})]$$
$$+ \mathbb{E}[f(y_t^*; \theta_{k\Delta+1}) - f(y_t^*; \theta_t)]$$
$$\leq 2\,\mathbb{E}[V^k].$$

By summing over $t \in \mathcal{T}_k$ and $k = 0, \ldots, \lceil T/\Delta \rceil - 1$, we obtain

$$\sum_{k=0}^{\lceil T/\Delta \rceil - 1} \sum_{t \in \mathcal{T}_k} \mathbb{E}[f(z_k^*; \theta_t) - f(y_t^*; \theta_t)] \leq 2\Delta\,\mathbb{E}[V_T]$$

Similar to the proof of Corollary 2, by applying the bounds above and $\Delta = \lceil \sqrt{2T/\mathbb{E}[V_T]} \rceil$, we obtain the desired bound on the expected dynamic regret of OGD for our setting, i.e.

$$\mathbb{E}[\text{Reg}(OGD)] \leq C_1 \sqrt{\mathbb{E}[V_T]T} \log(1 + \sqrt{T/\mathbb{E}[V_T]}) + \frac{h^2}{\alpha} \mathbb{E}[\|\boldsymbol{\delta}(\min(W,T))\|^2]$$

where $C_1$ is a constant factor defined in Theorem 2.

Consequently, by applying Theorem 4 and the bound on $\mathbb{E}[\|\boldsymbol{\delta}(W)\|^2]$ in (21), we have the following results.

$$\mathbb{E}[\text{Reg}(RHIG)] \leq C_2 \rho^W \sqrt{\mathbb{E}[V_T]T} \log(1 + \sqrt{T/\mathbb{E}[V_T]}) + \rho^W \frac{2h^2 L}{\alpha^2} \sum_{t=0}^{\min(W,T)-1} \|R_e\|(T-t)\|P(t)\|_F^2$$

$$+ \zeta \sum_{t=0}^{\min(W,T)-1} \|R_e\|(T-t)\|P(t)\|_F^2 \frac{\rho^t - \rho^W}{1-\rho}$$

$$\leq C_2 \rho^W \sqrt{\mathbb{E}[V_T]T} \log(1 + \sqrt{T/\mathbb{E}[V_T]}) + \zeta \sum_{t=0}^{\min(W,T)-1} \|R_e\|(T-t)\|P(t)\|_F^2 \frac{\rho^t}{1-\rho}$$

by $\frac{2h^2 L}{\alpha^2} - \frac{\zeta}{1-\rho} < 0$, where $C_2 = \frac{2L}{\alpha} C_1$.

## F.3  Proof of Theorem 5

The proof relies on the Hanson-Wright inequality in [39].[12]

**Proposition 1** (Hanson-Wright Inequality [39])**.** *Consider random Gaussian vector* $\boldsymbol{u} = (u_1, \dots, u_n)^\top$ *with* $u_i$ *i.i.d. following* $N(0,1)$*. There exists an absolute constant* $c > 0$,[13] *such that*

$$\mathbb{P}(\boldsymbol{u}^\top A \boldsymbol{u} \geq \mathbb{E}[\boldsymbol{u}^\top A \boldsymbol{u}] + b) \leq \exp\left(-c \min(\frac{b^2}{\|A\|_F^2}, \frac{b}{\|A\|})\right), \quad \forall b > 0$$

Now, we are ready for the proof. For any realization of the random vectors $\{e_t\}_{t=1}^T$, our regret bound in Corollary 1 still holds, i.e.

$$\text{Reg}(RHIG) \leq \rho^W \frac{2L}{\alpha} C_1 \sqrt{V_T T} \log(1 + \sqrt{T/V_T}) + \frac{2L}{\alpha} \frac{h^2}{\alpha} \rho^W \|\boldsymbol{\delta}(\min(W,T))\|^2$$

$$+ \sum_{k=1}^{\min(W,T)} \zeta \rho^{k-1} \|\boldsymbol{\delta}(k)\|^2 + \mathbb{1}_{(W>T)} \frac{\rho^T - \rho^W}{1-\rho} \zeta \|\boldsymbol{\delta}(T)\|^2$$

$$\leq \rho^W \frac{2L}{\alpha} C_1 T \log(2) + \frac{2L}{\alpha} \frac{h^2}{\alpha} \rho^W \|\boldsymbol{\delta}(\min(W,T))\|^2 + \sum_{k=1}^{\min(W,T)} \zeta \rho^{k-1} \|\boldsymbol{\delta}(k)\|^2$$

$$+ \mathbb{1}_{(W>T)} \frac{\rho^T - \rho^W}{1-\rho} \zeta \|\boldsymbol{\delta}(T)\|^2 =: R(W)$$

where we used the technical assumption that $V_T \leq T$. Notice that it can be verified that $\mathbb{E}[R(W)] \leq \mathbb{E}[\text{Regbdd}]$.

From (20) in the proof of Theorem 4, we have that $\boldsymbol{\delta}(k) = M_k e = M_k R_e^{1/2} \boldsymbol{u}$, where $\boldsymbol{u}$ is a standard Gaussian vector for $k \leq T$; and $\boldsymbol{\delta}(k) = M_T R_e^{1/2} \boldsymbol{u}$ for $k \geq T$.

When $W \leq T$, we have the following formula for $R(W)$.

$$R(W) = \rho^W \frac{2L}{\alpha} C_1 T \log(2) + \rho^W \frac{2L}{\alpha} \frac{h^2}{\alpha} \|\boldsymbol{\delta}(W)\|^2 + \zeta \sum_{k=1}^W \rho^{k-1} \|\boldsymbol{\delta}(k)\|^2$$

$$= \rho^W \frac{2L}{\alpha} C_1 T \log(2)$$

$$+ \boldsymbol{u}^\top \underbrace{(\rho^W \frac{2L}{\alpha} \frac{h^2}{\alpha} \boldsymbol{R_e}^{1/2} \boldsymbol{M}_W^\top \boldsymbol{M}_W \boldsymbol{R_e}^{1/2} + \zeta \sum_{k=1}^{W} \rho^{k-1} \boldsymbol{R_e}^{1/2} \boldsymbol{M}_k^\top \boldsymbol{M}_k \boldsymbol{R_e}^{1/2}) \, \boldsymbol{u}}_{\boldsymbol{A}_W}$$

We bound $\|\boldsymbol{A}_W\|_F$ below.

$$\|\boldsymbol{A}_W\|_F \leq \rho^W \frac{2L}{\alpha} \frac{h^2}{\alpha} \|\boldsymbol{R_e}^{1/2} \boldsymbol{M}_W^\top \boldsymbol{M}_W \boldsymbol{R_e}^{1/2}\|_F + \zeta \sum_{k=1}^{W} \rho^{k-1} \|\boldsymbol{R_e}^{1/2} \boldsymbol{M}_k^\top \boldsymbol{M}_k \boldsymbol{R_e}^{1/2}\|_F$$

$$\leq \rho^W \frac{2L}{\alpha} \frac{h^2}{\alpha} \|\boldsymbol{M}_W \boldsymbol{R_e}^{1/2}\|_F^2 + \zeta \sum_{k=1}^{W} \rho^{k-1} \|\boldsymbol{M}_k \boldsymbol{R_e}^{1/2}\|_F^2$$

$$= \rho^W \frac{2L}{\alpha} \frac{h^2}{\alpha} \operatorname{tr}(\boldsymbol{R_e}^{1/2} \boldsymbol{M}_W^\top \boldsymbol{M}_W \boldsymbol{R_e}^{1/2}) + \zeta \sum_{k=1}^{W} \rho^{k-1} \operatorname{tr}(\boldsymbol{R_e}^{1/2} \boldsymbol{M}_k^\top \boldsymbol{M}_k \boldsymbol{R_e}^{1/2})$$

$$= \rho^W \frac{2L}{\alpha} \frac{h^2}{\alpha} \operatorname{tr}(\boldsymbol{R_e} \boldsymbol{M}_W^\top \boldsymbol{M}_W) + \zeta \sum_{k=1}^{W} \rho^{k-1} \operatorname{tr}(\boldsymbol{R_e} \boldsymbol{M}_k^\top \boldsymbol{M}_k)$$

$$\leq \rho^W \frac{2L}{\alpha} \frac{h^2}{\alpha} \|R_e\| \operatorname{tr}(\boldsymbol{M}_W^\top \boldsymbol{M}_W) + \zeta \|R_e\| \sum_{k=1}^{W} \rho^{k-1} \operatorname{tr}(\boldsymbol{M}_k^\top \boldsymbol{M}_k)$$

$$= \rho^W \frac{2L}{\alpha} \frac{h^2}{\alpha} \|R_e\| \|\boldsymbol{M}_W\|_F^2 + \zeta \|R_e\| \sum_{k=1}^{W} \rho^{k-1} \|\boldsymbol{M}_k\|_F^2$$

$$= \rho^W \frac{2L}{\alpha} \frac{h^2}{\alpha} \|R_e\| \sum_{t=0}^{W-1} (T-t) \|P(t)\|_F^2 + \zeta \|R_e\| \sum_{k=1}^{W} \rho^{k-1} \sum_{t=0}^{k-1} (T-t) \|P(t)\|_F^2$$

$$= \rho^W \frac{2L}{\alpha} \frac{h^2}{\alpha} \|R_e\| \sum_{t=0}^{W-1} (T-t) \|P(t)\|_F^2 + \zeta \|R_e\| \sum_{t=0}^{W-1} \sum_{k=t+1}^{W} \rho^{k-1} (T-t) \|P(t)\|_F^2$$

$$= \rho^W \frac{2L}{\alpha} \frac{h^2}{\alpha} \|R_e\| \sum_{t=0}^{W-1} (T-t) \|P(t)\|_F^2 + \zeta \|R_e\| \sum_{t=0}^{W-1} \frac{\rho^t - \rho^W}{1-\rho} (T-t) \|P(t)\|_F^2$$

$$\leq \zeta \sum_{t=0}^{W-1} \|R_e\| (T-t) \|P(t)\|_F^2 \frac{\rho^t}{1-\rho}$$

where we used (21) and $\zeta = \frac{h^2}{\alpha} + \frac{h^2}{2L}$ and $\rho = 1 - \frac{\alpha}{4L}$.

When $W > T$, we have the following formula for the regret bound $R(W)$.

$$R(W) = \rho^W \frac{2L}{\alpha} C_1 T \log(2) + \frac{2h^2 L}{\alpha^2} \rho^W \|\boldsymbol{\delta}(T)\|^2 + \zeta \sum_{k=1}^{T} \rho^{k-1} \|\boldsymbol{\delta}(k)\|^2 + \zeta \|\boldsymbol{\delta}(T)\|^2 \frac{\rho^T - \rho^W}{1-\rho}$$

$$= \rho^W \frac{2L}{\alpha} C_1 T \log(2)$$

$$+ \boldsymbol{u}^\top \underbrace{\left( \left( \frac{2h^2 L}{\alpha^2} \rho^W + \zeta \frac{\rho^T - \rho^W}{1-\rho} \right) \boldsymbol{R_e}^{1/2} \boldsymbol{M}_T^\top \boldsymbol{M}_T \boldsymbol{R_e}^{1/2} + \zeta \sum_{k=1}^{T} \rho^{k-1} \boldsymbol{R_e}^{1/2} \boldsymbol{M}_k^\top \boldsymbol{M}_k \boldsymbol{R_e}^{1/2} \right) \boldsymbol{u}}_{\boldsymbol{A}_W}$$

Similarly, we bound $\|\boldsymbol{A}_W\|_F$ below.

$$\|\boldsymbol{A}_W\|_F \leq \left( \frac{2h^2 L}{\alpha^2} \rho^W + \zeta \frac{\rho^T - \rho^W}{1-\rho} \right) \|\boldsymbol{R_e}^{1/2} \boldsymbol{M}_T^\top \boldsymbol{M}_T \boldsymbol{R_e}^{1/2}\|_F + \zeta \sum_{k=1}^{T} \rho^{k-1} \|\boldsymbol{R_e}^{1/2} \boldsymbol{M}_k^\top \boldsymbol{M}_k \boldsymbol{R_e}^{1/2}\|_F$$

$$\leq \left( \frac{2h^2 L}{\alpha^2} \rho^W + \zeta \frac{\rho^T - \rho^W}{1 - \rho} \right) \|R_e\| \|\boldsymbol{M}_T\|_F^2 + \zeta \sum_{k=1}^{T} \rho^{k-1} \|R_e\| \|\boldsymbol{M}_k\|_F^2$$

$$\leq \left( \frac{2h^2 L}{\alpha^2} \rho^W + \zeta \frac{\rho^T - \rho^W}{1 - \rho} \right) \|R_e\| \sum_{t=0}^{T-1} (T-t) \|P(t)\|_F^2 + \zeta \sum_{k=1}^{T} \rho^{k-1} \|R_e\| \sum_{t=0}^{k-1} (T-t) \|P(t)\|_F^2$$

$$\leq \left( \frac{2h^2 L}{\alpha^2} \rho^W + \zeta \frac{\rho^T - \rho^W}{1 - \rho} \right) \|R_e\| \sum_{t=0}^{T-1} (T-t) \|P(t)\|_F^2 + \zeta \|R_e\| \sum_{t=0}^{T-1} \frac{\rho^t - \rho^T}{1 - \rho} (T-t) \|P(t)\|_F^2$$

$$\leq \zeta \sum_{t=0}^{T-1} \|R_e\| (T-t) \|P(t)\|_F^2 \frac{\rho^t}{1 - \rho}$$

In conclusion, for any $W \geq 1$, we have that $R(W) = \rho^W \frac{2L}{\alpha} C_1 T \log(2) + \boldsymbol{u}^\top \boldsymbol{A}_W \boldsymbol{u}$, and $\|\boldsymbol{A}_W\|_F \leq \zeta \sum_{t=0}^{\min(W,T)-1} \|R_e\| (T-t) \|P(t)\|_F^2 \frac{\rho^t}{1-\rho}$. Further, we have $\|\boldsymbol{A}_W\| \leq \|\boldsymbol{A}_W\|_F$. Therefore, by Proposition 1, we prove the concentration bound below. For any $b > 0$,

$$\mathbb{P}(\text{Reg}(RHIG) \geq \mathbb{E}[\text{Regbdd}] + b) \leq \mathbb{P}(R(W) \geq \mathbb{E}[\text{Regbdd}] + b)$$
$$\leq \mathbb{P}(R(W) \geq \mathbb{E}[R(W)] + b)$$
$$= \mathbb{P}(\boldsymbol{u}^\top \boldsymbol{A}_W \boldsymbol{u} \geq \mathbb{E}[\boldsymbol{u}^\top \boldsymbol{A}_W \boldsymbol{u}] + b)$$
$$\leq \exp\left( -c \min\left( \frac{b^2}{K^2}, \frac{b}{K} \right) \right)$$

where $K = \zeta \sum_{t=0}^{\min(T,W)-1} \|R_e\| (T-t) \|P(t)\|_F^2 \frac{\rho^t}{1-\rho}$.

## G More details of the numerical experiments

**(i) The high-level planning problem.** The parameters are: $e_t \sim N(0,1)$ i.i.d., $T = 20$, $\alpha = 1$, $\beta = 0.5$, $x_0 = 10$, $a = 4$, $\omega = 0.5$, $\eta = 0.5$, $\xi_t = 1$, CHC's commitment level $v = 3$. The regret is averaged over 200 iterations.

**(ii) The physical tracking problem.** Consider the second-order system
$$\ddot{x} = k_1 u + g + k_2$$
where $x$ is altitude, $\dot{x}$ is velocity, $\ddot{x}$ is acceleration, etc.

Consider a discrete-time version of the system above as
$$\frac{x_{t+1} - 2x_t + x_{t-1}}{\Delta^2} = k_1 u_t - g + k_2$$

which is equivalent to
$$u_t = \frac{1}{k_1} \left( \frac{x_{t+1} - 2x_t + x_{t-1}}{\Delta^2} - (-g + k_2) \right).$$

Consider a cost function at stage $t$ as
$$\frac{\alpha}{2} (x_t - \theta_t)^2 + \frac{\beta}{2} u_t^2.$$

We can write the cost function in terms of $x_t$, that is,
$$\frac{\alpha}{2} (x_t - \theta_t)^2 + \frac{\beta}{2} \frac{1}{k_1^2} \left( \frac{x_{t+1} - 2x_t + x_{t-1}}{\Delta^2} - (-g + k_2) \right)^2.$$

Notice that the switching cost is not $d(x_t, x_{t-1})$ but $d(x_{t+1}, x_t, x_{t-1})$, but we still have the local coupling property of the gradients and we can still apply RHIG.

The experiment parameters are provided below. Consider horizon 10 seconds and time discretization $\Delta = 0.1$s. Let $k_1 = 1$, $k_2 = 1$, $\alpha = 1$, $\beta = 1 \times 10^{-5}$, $x_0 = 1$m, $g = 9.8$m/s$^2$. Let $e_t \sim N(0, 0.5^2)$ i.i.d. for all $t$. Consider $d_t = 0.9 \sin(0.2t) + 1$ before $t \leq 5.6$s and $d_t = 0.3 \sin(0.2t) + 1$ afterwards. Let $\gamma = 0.6$, $\xi_t = 1$, $\eta = 1/L$ and $L \approx 2.6$.