[Reviews · NeurIPS 2020]

Review 1

Summary and Contributions: This paper considers online convex optimization with switching costs and studies how to reduce the impact of prediction errors on performance. A new algorithm, Receding Horizon Inexact Gradient (RHIG) is presented and regret bounds are derived under a general model of prediction errors. Additionally, numerical examples in the context of a quadrotor tracking problem are included.

Strengths: This is a theoretical paper that makes a clear contribution to a growing literature. It is relevant to NeurIPS and bridges control and online learning. The topic of predictions in online optimization has been studied a lot in the past few years at the intersection of control and learning, and typical papers use very idealistic models without realistic assumptions about errors. This paper considers a general error model and is still able to derive strong analytic results.

Weaknesses: The main contribution of the paper is the theoretical results and analysis and the paper provides more general results in an more realistic model than the prior literature. However, it was difficult for me to understand the technical novelty in the proofs. In particular, the algorithmic idea is related to prior work that mixes RHS and gradient algorithms, e.g., [12]. I am interested to have the authors clarify what new ideas/techniques in the proof enable the new results?

Correctness: I have verified the provided proofs.

Clarity: The paper is clear, but some details in the presentation can be improved. -- It is useful to highlight more clearly the novelty of the techniques used in the analysis. -- More discussion of the differentiation between RHIG and other algorithms in the literature would be helpful. There is a growing zoo of related algorithms and understanding what is crucial for the noisy predictions is valuable.

Relation to Prior Work: The relationship to prior work is discussed clearly, albeit concisely. If space allows I would suggest including additional context about the related work.

Reproducibility: Yes

Additional Feedback: Thank you for your author response. I have read it and it clarified a few issues for me around the prediction model.


Review 2

Summary and Contributions: The paper studies the case of strongly-convex online optimization with switching cost and where the learner has access to multi-step ahead predictions. Namely, at each round, the learner receives inaccurate prediction of all future losses. The paper proceeds in giving an algorithm (RHIG) that provides bounded policy regret (against any non-stationary benchmark) that scales with the variation of the losses. The algorithm sets a parameter W of how many lookahead steps it utilizes and the regret bound shows that W trades off between the variation of the losses and the accuracy of the predictions. In particular, the regret bound decays exponentially in W. The authors conclude by evaluating their algorithm on a synthetic experiment.

Strengths: The paper introduces a new model for online optimization.

Weaknesses: The main body of the paper does not include any explanation about the regret bound. The authors never explain why the regret bound decays exponentially in W nor give any other intuition on the algorithm.

Correctness: Given that there is no explanation of the analysis in the main body of the paper, that makes it difficult to verify the correctness of the proofs.

Clarity: I found the paper to be written okay for the most part, though it obscures much of the technical details (see 'weaknesses').

Relation to Prior Work: This work extends the RHGD algorithm of Li et al. to the case where future predictions are inaccurate.

Reproducibility: Yes

Additional Feedback: Here are some references that I think were missed: 1. "Online Learning for Adversaries with Memory: Price of Past Mistakes", Anava et al. See Section 6. This work addresses a model similar to the authors' stochastic model in the context of multi-step prediction. 2. "Online Learning with Predictable Sequences", Rakhlin and Sridharan. This work addresses the case where the learner has information on the next time step and gives a regret bound that scales with the accuracy of the predictions. -- After rebuttal -- The rebuttal did not alleviate concerns that I had with the papers writing and the readers' ability to validate the main results. Specifically: the explanation of the proof and the main theorem statement, and the lacking comparison to previous work. I am therefore keeping my score.


Review 3

Summary and Contributions: This paper considers smoothed online convex optimization (SOCO) with multi-step-ahead predictions, where in each round t the learner can get access to a series of noised predictions of the stage cost functions in the following rounds, and the prediction errors can be either fixed or stochastic. The authors propose a variant of the RHGD algorithm to solve this problem. Theoretical analysis shows that the dynamic regret of the proposed method is related to the function variations and perdition errors. Numerical experiments demonstrate that the proposed method outperforms previous algorithms when the prediction errors are large.

Strengths: 1. The problem is well-motivated. Previous work on SOCO [14] assumes that the in each round t the stage functions in the following W rounds can be precisely revealed to the learner, which is unrealistic and hinders their applications to many domains. In this paper, the authors successfully relaxed this assumption by proposing a new model where the learner can only get access to a noised version of the future stage cost functions. 2. The authors successfully proposed a variant of RHGD to solve this problem. They provide a dynamic regret bound for the proposed algorithm. Moreover, the numerical experiments show that the proposed method outperforms previous algorithms when the prediction errors are large.

Weaknesses: Significance: 1. This paper assumes that all the stage cost functions share the same form, and the function form is known to the learner before the whole iteration process, which seems to be a very limited setting. In previous work (e.g. [14]), the type of loss functions can be different in different rounds, and in round t only the types of the stage cost functions in the following W rounds are revealed to the learner. 2. This paper and [14] assume the stage cost functions are both smooth and strongly convex, which is also a very restrictive assumption. It would be nice if the authors could give more examples of loss functions (other than quadratic functions) which satisfy these assumptions. 3. I think the theoretical results are not very clear. For example, in Corollary 1, it is hard to tell what is the exact order of the regret bound. It would be better if the authors could give more specific examples of delta, in which cases the regret bounds are sublinear (with respect to V_T and T). Novelty: It seems to me that both the proposed algorithm and its theoretical analysis are largely based on those in [14]. The main theoretical challenge is how to deal with the prediction errors, and according to Lemma 4 and Lemma 5, it seems that the error terms can be handled pretty straightforwardly. It would be nice if the authors could highlight the main theoretical difficulties in the main paper. -----------------------Post Rebuttal--------------------------- The authors cleared most of my concerns in the retuttal. I am happy to raise my score.

Correctness: I have read the main paper and made high level checks of the proofs, and I didn’t find any significant errors.

Clarity: The paper is generally well-written and structured clearly. However, as mentioned above, I think the theoretical results (dynamic regret bounds) are not very clear. Perheaps the authors can add more discuessions and give more examples on this problem.

Relation to Prior Work: The relation to prior work is clearly discuessed in general. However, it seems that the proof of Theorem 1 is largely based on the proof of Theorem 1 in [14]. It would be better if the authors can make it more clear in the main paper and the appendix.

Reproducibility: Yes

Additional Feedback: In Corollary 3, for small W (say W=1), is the second term linear with respect to T?


Review 4

Summary and Contributions: The paper introduce a general online convex optimization algorithmic framework that can handle additional switching costs. The algorithm leverage long-term prediction to improve online performance. The tradeoff between accuracy that longer-term prediction provides and the larger error of lone-term prediction is crucial, and the paper incorporate it as a parameter to tune. Both theoretical regret bounds and empirical demonstration are provided to support the algorithm.

Strengths: Online convex optimization is extensively studied in the literature. Adding a switch cost is less studied but has a wide range of application. I think this paper provides a general algorithmic framework to understand this problem better.

Weaknesses: The key weakness of this paper is the comparison to other possible approaches. The very first baseline the paper need to compare is RHGD algorithm, which they does not explicitly compare their regret bound and empirical result. Since RHIG is the natural extension of RHGD, I think a direct comparison to show the benefit is important, otherwise the novelty is not enough. Another line of research is reinforcement learning, which we can take the switching cost naturally as the reward given after applying action to the current state. I don't find a detailed discussion of this line of work in the paper.

Correctness: I take a quick look at the theory and seems the claim the proof are sound. The numerical experiment methodology is also standard.

Clarity: I enjoy reading most of the sections, the only confusing part for me the difference between RHIG and RHGD, which I only find a few line (126-128) to describe their difference. I am not fully convinced that their difference is so significant without showing the exact comparison both theoretically and empirically.

Relation to Prior Work: As I mentioned in the discussion on weakness part, I think the paper need to add more detailed discussion with RHGD and reinforcement learning.

Reproducibility: Yes

Additional Feedback:

[Author Response · NeurIPS 2020]

We thank the reviewers for their constructive comments. We will incorporate the discussions below in the revision.

**R1-Q1 & R3-Q4: technical challenges and novelty.** We apologize for the confusion. One of the main challenges
we encountered was *how to model/represent the (multi-step-ahead) prediction errors*. The model should be general
enough to include various prediction methods, yet clean enough to allow for insightful theoretical analysis. After many
attempts, we eventually decide to represent prediction errors by how many steps ahead this prediction is made. This
representation is general and can include most existing models in literature [1,15,25]. Further, this representation clearly
suggests how each prediction errors affect the algorithm updates (see e.g. (6)), providing insightful understanding.

Next, we discuss the proof novelty. Similar to RHGD in [12] and [14], we design our RHIG in such a way that the
online regret analysis can be converted to offline gradient-based algorithms' convergence analysis. However, our RHIG
is converted to offline *inexact/stochastic* gradient descent, where the gradient errors are from prediction errors; while
[12,14] only deal with the *exact* gradient, which is much simpler to analyze. To handle the prediction/gradient errors,
we first establish the almost-strong-convexity and almost-smoothness inequalities in Lemma 4 & 5 as mentioned by R3.
In addition, we conduct nontrivial manipulation of the inequalities and carefully choose the stepsizes and parameters so
that the accumulated errors do not explode. Our procedures are different from the classic literature since our gradient
errors are time-varying, not uniformly bounded, and correlated, while most literature assumes uniformly bounded and/or
i.i.d. gradient errors. Moreover, we provide a concentration bound for the regret, which is not addressed in [12,14].

Finally, we note that existing papers on SOCO with noisy multi-step predictions (e.g. [1,15]) only provide regret bounds
for *optimization-based* algorithms, but our paper studies a *gradient-based* algorithm, which is more applicable under
computational time constraints, e.g. in robotics, connected vehicles and other large transportation systems, etc.

**R1-Q2 & R2-Q2: algorithm comparison and intuitions on how we deal with noisy predictions.** Good question.
Firstly, most existing methods, e.g. AFHC, CHC, are optimization-based, while our RHIG is based on gradient descent,
which is known to be more robust to errors. Secondly, RHIG implicitly reduces the impact of the (poorer-quality)
long-term predictions and focuses more on the (better) short-term ones by using long-term predictions in the first several
updates and then using short-term ones in later updates to refine the decisions; while AFHC and CHC treat predictions
more equally by taking averages of the optimal solutions computed by both long-term and short-term predictions.

**R2-Q1: regret bound explanation.** We apologize for the confusion. Let $E_k = \|\boldsymbol{\delta}(k)\|^2$ denote the $k$-step-ahead
prediction error, then the regret bound in Corollary 1 is $\tilde{O}(\rho^W \sqrt{V_T T} + \sum_{k=1}^{\min(W,T)} \rho^k E_k)$. The first term decays
exponentially with $W$, suggesting that RHIG reduces the impact of the variation $V_T$ exponentially with $W$. In the
second term, $E_k$'s coefficient $\rho^k$ decays exponentially with $k$, suggesting that *the impact of the $k$-step-ahead prediction*
*error decays exponentially with $k$*. This benefits the performance since long-term predictions are usually much poorer
than short-term ones. **R2-Q3: proof explanation.** Please kindly refer to the second paragraph of R1-Q1 & R3-Q4.

**R3-Q1: parametric costs.** Good question. Section 4 (the deterministic case) can be generalized to varying function
types as in [14]. However, the tricky problem is *how to model the stochastic prediction errors*. There are usually three
types of correlation between the prediction errors, i.e. between the errors of (i) $\nabla f_{t|\tau}(x_1)$ and $\nabla f_{t|\tau}(x_2)$, (ii) $\nabla f_{t|\tau_1}(x)$
and $\nabla f_{t|\tau_2}(x)$, (iii) $\nabla f_{t_1|\tau}(x)$ and $\nabla f_{t_2|\tau}(x)$. Due to the complicated correlation, we decide to adopt the stochastic
model in [1,15], which uses parametric functions $f(x;\theta)$ to capture the type-i correlation, and uses equation (9) to
model the type-ii and type-iii. **R3-Q2: strongly convex and smooth costs.** Consider $l_2$-regularized logistic regression:
$\min_x f(x; \{z_i, y_i\}_{i=1}^n) = -\sum_{i=1}^n \log(1+\exp(-y_i\langle x, z_i\rangle))/n + \lambda\|x\|^2/2$, where $x$ is the regressor, $(z_i, y_i)_{i=1}^n$ denotes
the samples, and $\lambda$ is $l_2$ regularizer's coefficient. The loss function is strongly convex and smooth. (see [2] for more
examples.) **R3-Q3: regret bound's order.** Please refer to R2-Q1 for the regret bound in the big-$O$ notation. The
prediction errors $E_k$ can be either larger or smaller than $V_T$ as mentioned in [20]. When $V_T = o(T)$ and $E_k = o(T)$
for $k \leq W$, the regret bound is $o(T)$. As a simple example of sublinear regrets, consider $\theta_{t-1}$ as the prediction of $\theta_{t+k}$
($k \geq 0$) at time $t$, then $E_k = O(V_T)$ under proper assumptions, so when $V_T = o(T)$, the regret is $o(T)$. **R3-Q5: Cor.**
**3 is $O(T)$ when $W = 1$?** Yes. This is consistent with classic results in [1,15]. Intuitively, the stochastic model (9)
indicates that the total variance of the prediction errors is $O(T)$, so the regret is at least $O(T)$ by the perturbation theory.

**R4-Q1: comparison with RHGD.** RHGD requires *accurate $W$-stage predictions*; while our RHIG can handle
*inaccurate predictions*, which is more realistic and thus our regret bounds are more general and meaningful. For
more technical discussion, please kindly refer to R1-Q1 & R3-Q4. Numerically, we observe better performance by
RHIG since the carefully chosen stepsizes and parameters effectively reduce the error accumulation, while RHGD's
performance is seriously downgraded by the prediction errors. **R4-Q2: comparison with RL.** Good question. This
work and RL focus on different issues. Classical RL focuses on *unknown* systems in *stationary* environment, while this
paper considers *time-varying/nonstationary* environment with *noisy future predictions* and a *known system*. Though
one can apply RL algorithms to time-varying environment anyway, the online performance could be very undesirable
without special treatment of the nonstationary environment. Our work addresses two under-explored issues in RL: *how*
*to use the future predictions to improve the performance in nonstationary environment?* and *how to characterize the*
*impact of the prediction errors?* Combining this work and RL is an exciting direction to promote RL's performance.

[Meta-Review · NeurIPS 2020]

The paper considers online convex optimization with time-varying stage costs and additional switching costs, with long-term predictions incorporated to improve the online performance. A new algorithm, Receding Horizon Inexact Gradient is given, with guarantees on its regret under a general model of prediction errors. The reviewers generally liked the paper, and appreciated a well-motivated problem, a clear contribution to the theory and rigorous analysis of the proposed algorithm, as well as a general prediction error model (as opposed to idealistic models used in the past work). On the other hand, there were several issues raised by the reviewers: lack of explanation for the regret bound and the analysis in the main body of the paper, lacking comparison to the past work, assumption that all stage functions share the same form, smoothness and strong convexity of the stage costs. Most of these concerns appear to be clarified in the rebuttal.